# 🤖 EMBODIEDBENCH: Comprehensive Benchmarking Multi-modal Large Language Models for Vision-Driven Embodied Agents

Rui Yang [1] [*]   Hanyang Chen [1] [*]   Junyu Zhang [1] [*]   Mark Zhao [3] [‡] [*]
Cheng Qian [1]   Kangrui Wang [2]   Qineng Wang [2]   Teja Venkat Koripella [1]   Marziyeh Movahedi [4] [‡]
Manling Li [2]   Heng Ji [1]   Huan Zhang [1] [†]   Tong Zhang [1] [†]

## Abstract

Leveraging Multi-modal Large Language Models (MLLMs) to create embodied agents offers a promising avenue for tackling real-world tasks. While language-centric embodied agents have garnered substantial attention, MLLM-based embodied agents remain underexplored due to the lack of comprehensive evaluation frameworks. To bridge this gap, we introduce EMBODIEDBENCH, an extensive benchmark designed to evaluate vision-driven embodied agents. EMBODIEDBENCH features: (1) a diverse set of 1,128 testing tasks across four environments, ranging from high-level semantic tasks (e.g., household) to low-level tasks involving atomic actions (e.g., navigation and manipulation); and (2) six meticulously curated subsets evaluating essential agent capabilities like commonsense reasoning, complex instruction understanding, spatial awareness, visual perception, and long-term planning. Through extensive experiments, we evaluated 24 leading proprietary and open-source MLLMs within EMBODIEDBENCH. Our findings reveal that: MLLMs excel at high-level tasks but struggle with low-level manipulation, with the best model, GPT-4o, scoring only 28.9% on average. EMBODIEDBENCH provides a multifaceted standardized evaluation platform that not only highlights existing challenges but also offers valuable insights to advance MLLM-based embodied agents. Our code and dataset are available at https://embodiedbench.github.io.

[*]Equal contribution [†]Equal advising [1]University of Illinois Urbana-Champaign [2]Northwestern University. [3]University of Toronto. [4]Toyota Technological Institute at Chicago. [‡]Work done during internship at UIUC. Correspondence to: Rui Yang <ry21@illinois.edu>, Huan Zhang <huan@huan-zhang.com>, Tong Zhang <tongzhang@tongzhang-ml.org>.

*Proceedings of the 42$^{nd}$ International Conference on Machine Learning*, Vancouver, Canada. PMLR 267, 2025. Copyright 2025 by the author(s).

## 1. Introduction

Developing embodied agents capable of solving complex tasks in real world remains a significant challenge (Durante et al., 2024). Recent advancements in foundation models, including Large Language Models (LLMs) (Brown et al., 2020; Achiam et al., 2023; Touvron et al., 2023; Yang et al., 2024a) and Multimodal Large Language Models (MLLMs) (OpenAI, 2024a; Reid et al., 2024; Liu et al., 2024a; Wang et al., 2024; Chen et al., 2023c; 2025), have unlocked unprecedented potential toward this goal. These models, trained on extensive internet-scale datasets, demonstrate exceptional proficiency in understanding human knowledge and performing human-like reasoning. Based on these capabilities, researchers can now design intelligent agents that use off-the-shelf foundation models to solve complex tasks through interaction with environments (Huang et al., 2022a;b; 2023c; Ahn et al., 2022; Song et al., 2023; Singh et al., 2023; Liang et al., 2023; Qian et al., 2024).

Given the multitude of proposed algorithms, there is a pressing need for standardized and automated evaluation frameworks to enable comprehensive assessment and comparison. To address this need, several initiatives have been exploring LLM-based embodied agent evaluation (Liu et al., 2023b; Choi et al., 2024; Li et al., 2024b). While these efforts significantly contribute to understanding LLM-based agent design, the evaluation of MLLM embodied agents remains underexplored, posing a challenge for creating more versatile agents. VisualAgentBench (Liu et al., 2024e) represents the first benchmark for evaluating MLLM agents, covering embodied tasks such as household and Minecraft. However, its limited scope, focusing exclusively on high-level planning, leaves critical questions unanswered, such as *the role of vision in embodied tasks and the performance of MLLM agents in low-level tasks like navigation and manipulation.*

To address these questions, we introduce EMBODIED-BENCH, a comprehensive benchmark comprising 1,128 testing instances across four environments. EMBODIEDBENCH is designed with two key features that set it apart from existing benchmarks: **1. Diverse tasks with hierarchical action levels.** Among the four environments, EB-ALFRED and EB-

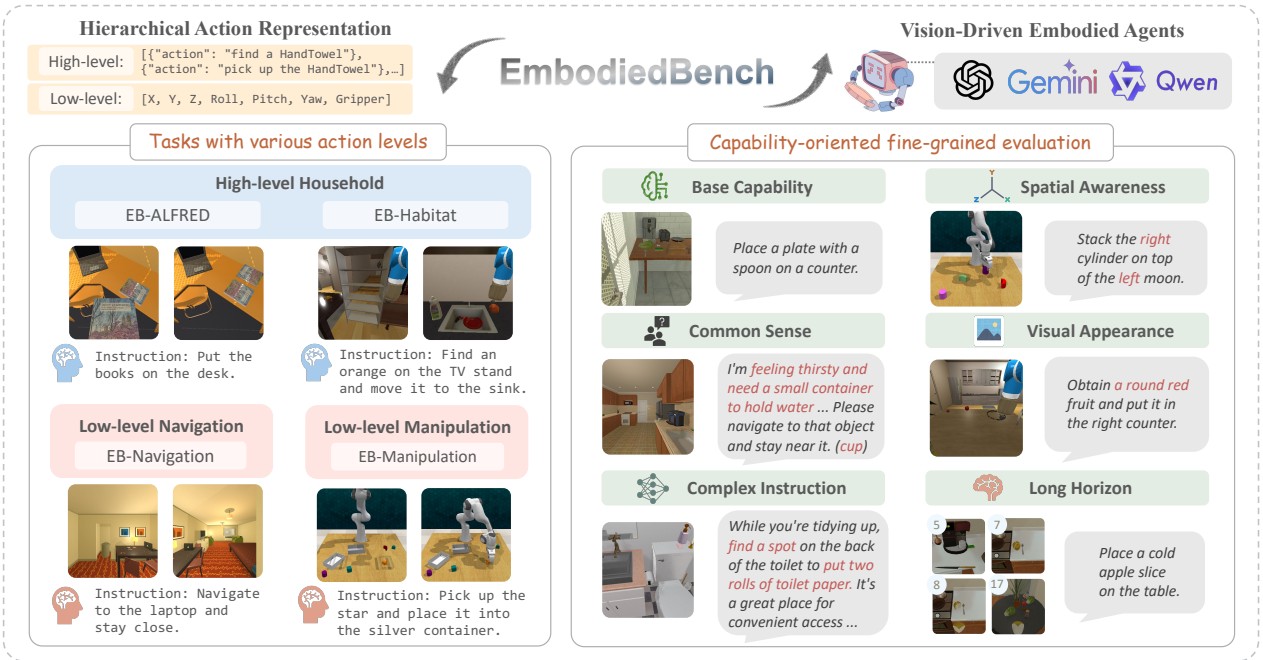

Figure 1. Overview of EMBODIEDBENCH. Two key features of our benchmark: various action levels and capability-oriented evaluation.

Habitat focus on high-level task decomposition and planning (e.g., "put a book on the desk"), while EB-Navigation and EB-Manipulation demand planning with low-level actions (e.g., translational/rotational control) and require precise perception and spatial reasoning. **2. Capability-oriented evaluation.** Unlike previous benchmarks that primarily emphasize overall accuracy (Liu et al., 2023b; Choi et al., 2024; Liu et al., 2024e) or module-specific performance (Li et al., 2024b), EMBODIEDBENCH introduces a fine-grained evaluation framework that assesses six critical capabilities of embodied agents, including basic task solving, common-sense reasoning, complex instruction understanding, spatial awareness, visual perception, and long-horizon planning.

To facilitate the evaluation of MLLMs as embodied agents, we design a unified agent framework that integrates ego-centric visual perception, few-shot in-context examples, interaction history, and environment feedback for decision-making. This powerful framework can unlock the full potential of current off-the-shelf MLLMs and tackle both high-level and low-level tasks effectively. Based on EMBODIEDBENCH and our agent pipeline, we evaluate 24 leading closed-source MLLMs (e.g., GPT-4o, Gemini, Claude-3.7, and Qwen-VL-Max) and 7B–90B open-source models (e.g., Llama-3.2 Vision (Meta, 2024), InternVL3 (Zhu et al., 2025), Qwen2.5-VL (Bai et al., 2025), and Gemma-3 (Team et al., 2025)). Our evaluation yields three key findings: (1) While MLLMs excel at high-level tasks, they struggle with low-level manipulation. (2) Long-horizon planning emerges as the most challenging subset. (3) Vision input is crucial for low-level tasks, with performance degrading by 40%–70% when removed, whereas its impact on high-level tasks is minimal. Additionally, our ablation studies provide practical insights into MLLM agent design, particularly regarding image resolution, multi-step image input, and visual in-context learning.

Our contributions are threefold: (1) proposing a comprehensive benchmark suite for evaluating MLLM-based embodied agents with different action levels and fine-grained capability-oriented subsets, (2) the development of an efficient MLLM agent framework, (3) conducting extensive evaluations and ablation studies of leading MLLMs, providing valuable insights for vision-driven agent design.

## 2. Related Work

In embodied agent research, LLMs are primarily used to support high-level planning (Ahn et al., 2022; Huang et al., 2022a;b; Yao et al., 2023; Huang et al., 2023d; Rana et al., 2023; Chen et al., 2023a; Gao et al., 2024b). MLLMs are then integrated for perception-related tasks (Chen et al., 2023b; Wang et al., 2023d; Gao et al., 2024b). Beyond perception, MLLMs also contribute to decision-making, either by directly generating actions in an end-to-end manner (Shridhar et al., 2022; Driess et al., 2023; Du et al., 2023; Mu et al., 2024) or by producing code to develop policy or value functions (Liang et al., 2023; Huang et al., 2023c).

As this field rapidly evolves, a variety of simulators (Kolve et al., 2017; Shridhar et al., 2020a; Xiang et al., 2020; Li et al., 2021; 2023) and evaluation benchmarks (Shridhar et al., 2020b;a; James et al., 2020; Zheng et al., 2022; Szot

*Table 1.* Comparison with related benchmarks. EMBODIEDBENCH is a multi-domain benchmark including household, manipulation, and navigation tasks. "Fine-grained" indicates a multi-dimensional evaluation approach rather than an overall accuracy. [1]AgentBench and VisualAgentBench include domains such as household, games, and Web. [2]VLABench is originally used for evaluating VLA models.

| Benchmark | Category | Action Level | #Env. | #Test Tasks | Multimodal | Fine-grained | LLM/VLM Support |
|---|---|---|---|---|---|---|---|
| ALFWorld (Shridhar et al., 2020b) | Household | High | 1 | 274 | ✗ | ✗ | ✗ |
| Alfred (Shridhar et al., 2020a) | Household | High | 1 | 3062 | ✓ | ✗ | ✗ |
| VLMbench (Zheng et al., 2022) | Manipulation | Low | 1 | 4760 | ✓ | ✗ | ✗ |
| Behavior-1K (Li et al., 2023) | Household | High | 1 | 1000 | ✓ | ✗ | ✗ |
| Language Rearrangement (Szot et al., 2023) | Household | High | 1 | 1000 | ✓ | ✓ | ✗ |
| GOAT-bench (Khanna et al., 2024) | Navigation | Low | 1 | 3919 | ✓ | ✗ | ✗ |
| AgentBench (Liu et al., 2023b) | Multi-domain[1] | High | 8 | 1091 | ✗ | ✗ | ✓ |
| Lota-bench (Choi et al., 2024) | Household | High | 2 | 308 | ✗ | ✗ | ✓ |
| VisualAgentBench (Liu et al., 2024e) | Multi-domain[1] | High | 5 | 746 | ✓ | ✗ | ✓ |
| Embodied Agent Interface (Li et al., 2024b) | Household | High | 2 | 438 | ✗ | ✓ | ✓ |
| VLABench (Zhang et al., 2024a) | Manipulation | Low[2] | 1 | 100 | ✓ | ✓ | ✓ |
| **EMBODIEDBENCH (ours)** | Multi-domain | High & Low | 4 | 1128 | ✓ | ✓ | ✓ |

et al., 2023; Liu et al., 2023b; 2024e; Choi et al., 2024; Li et al., 2024b; Zhang et al., 2024a; Cheng et al., 2025) have emerged. Table 1 provides a comprehensive comparison with existing works, highlighting how EMBODIEDBENCH sets itself apart from prior works in several aspects. More related works are listed in Appendix A.

## 3. Problem Formulation

**Definition of Action Levels.** In embodied agent research, actions can be systematically classified into hierarchical levels based on their executability in robotic systems (Ma et al., 2024b; Belkhale et al., 2024). **Low-level actions** correspond to atomic commands directly executable by robots, defined as operations that specify translational or rotational displacements. For instance, a robotic arm's action is often parameterized as a 7-dimensional vector: $a = [X, Y, Z, \text{Roll}, \text{Pitch}, \text{Yaw}, \text{Gripper}]$, where $(X, Y, Z)$ denote incremental translational displacements, $(\text{Roll}, \text{Pitch}, \text{Yaw})$ represent rotational deltas in Euler angles, and $\text{Gripper}$ encodes the binary open/closed state of the end-effector. Similarly, commands like "move forward 0.1 m" qualify as low-level actions, as they map unambiguously to kinematic transformations. In contrast, **high-level actions** can be decomposed into sequences of low-level primitives. Formally, a high-level action is defined as $a^h = [a_1, a_2, \ldots, a_n]$, where each $a_i$ is a low-level executable primitive. For example, executing "find a HandTowel" might involve iterating through low-level behaviors: rotating certain degrees, scanning for the target, and moving towards it.

**Vision-driven Agents.** Vision-driven agents are autonomous systems that make sequential decisions based on visual perception and language instructions. This problem can be formally modeled as a Partially Observable Markov Decision Process (POMDP) augmented with language instructions, defined by the tuple $(\mathcal{S}, \mathcal{A}, \Omega, \mathcal{T}, \mathcal{O}, L, \mathcal{R})$. Here, $\mathcal{S}$ is the complete state space unobservable to the agent; $\mathcal{A}$ is the space of high-level or low-level actions for the

agents; $\Omega$ is the visual perception space, where each observation $I_t \in \Omega$ corresponds to an image frame at time $t$; $\mathcal{T}$ is the transition dynamics; $\mathcal{O}$ relates the underlying states to the agent's visual observations; $L$ is the language instruction that specifies the desired goal; $\mathcal{R}$ evaluates task completion given the language instruction $L$: $r_t =$
$$\begin{cases} 1 & \text{if } s_t \models L \text{ (instruction achieved)} \\ 0 & \text{otherwise} \end{cases}$$
. At timestep $t$, the agent maintains a history $h_t = (I_0, a_0, ..., I_{t-1}, a_{t-1}, I_t)$ and selects actions through a policy $\pi(a_t|L, h_t)$. The objective is to maximize the probability of task success: $\max_\pi \mathbb{E}[r_\tau]$, where $\tau$ is the terminal timestep—either when the task is successfully completed ($s_\tau \models L$) or when the maximum horizon is reached.

## 4. EmbodiedBench

To thoroughly assess MLLMs as embodied agents across various action levels and capabilities, we introduce EM-BODIEDBENCH, a benchmark comprising four environments: EB-ALFRED, EB-Habitat, EB-Navigation, and EB-Manipulation. To evaluate six core embodied agents' capabilities, we developed new datasets and enhanced existing simulators to support comprehensive assessments. Below is an overview of the four benchmark tasks, with further details available in Appendix C.

### 4.1. High-level and Low-level Tasks

**EB-ALFRED.** We develop EB-ALFRED based on the AL-FRED dataset (Shridhar et al., 2020a) and the AI2-THOR simulator (Kolve et al., 2017). Our simulator is based on Lota-Bench's implementation (Choi et al., 2024) for 8 high-level skill types: "pick up", "open", "close", "turn on", "turn off", "slice", "put down", and "find", each customizable with specific objects, for example, "find an apple". The simulator provides an egocentric view as observation, along with textual feedback on action validity and possible failure reasons. Despite its strengths, Lota-Bench's simulator has several limitations, which we outline in Appendix C.1. To

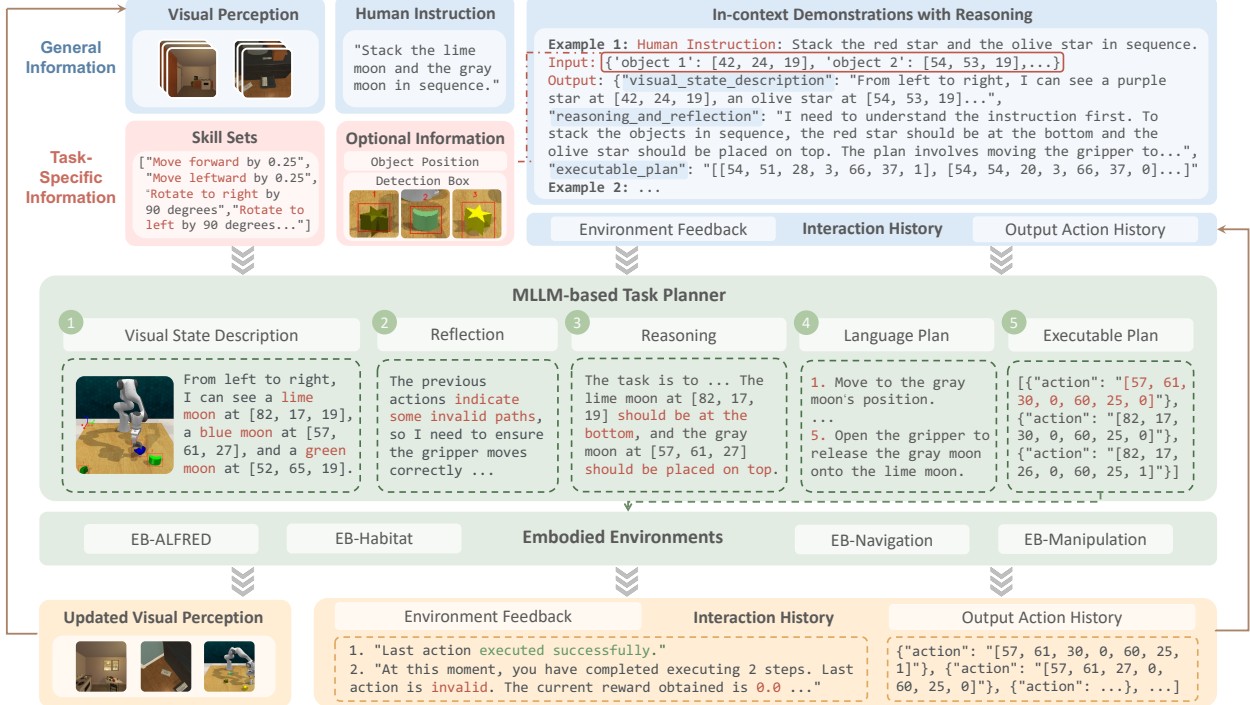

Figure 2. The vision-driven agent pipeline used in EMBODIEDBENCH. This pipeline serves as a robust framework for processing multimodal inputs, reflection and reasoning, and generating executable plans. For detailed descriptions, refer to Section 4.3.

enhance the simulation, we introduced key improvements, such as support for multiple instances of the same object type, allowing us to cover all task types in ALFRED. Additionally, we streamlined the action space by merging "put down" actions into a single action, since only one object can be held at a time. Due to the varying number of objects in ALFRED, the action space of EB-ALFRED is dynamic, ranging from 171 to 298 actions. Furthermore, we manually corrected simulator errors and refined instruction quality, ensuring more accurate action execution and improved task solvability. These enhancements make EB-ALFRED a high-quality benchmark for evaluating embodied agents.

**EB-Habitat.** EB-Habitat is built upon the Language Rearrangement benchmark (Szot et al., 2023), featuring 282 diverse language instruction templates. It leverages the Habitat 2.0 simulator (Szot et al., 2021) and focuses on planning and executing 70 high-level skills to achieve user-defined goals. These skills fall into five categories: "navigation", "pick", "place", "open", and "close", with each skill parameterized by a set of objects. Unlike ALFRED, which permits navigation to any object, EB-Habitat restricts navigation to receptacle-type objects, requiring robots to visit multiple locations to find desired items. With its wide variety of language instructions and unique navigation constraints, EB-Habitat serves as a valuable complement to EB-ALFRED.

**EB-Navigation.** EB-Navigation is an evaluation suite based on AI2-THOR (Kolve et al., 2017), designed to assess em-

bodied agents' navigation abilities. Each unique navigation task is primarily defined by: (1) *initial Robot Pose*, (2) *target object information*, and (3) *language instruction* that specifies which target object to locate, such as "navigate to the laptop". The robot can only rely on visual observations and textual feedback, without direct positioning data, to navigate to the target object. Success is defined as reaching within a specified distance of the target. The action space includes 8 low-level actions: (1) Move forward/backward/left/right by $\Delta x$. (2) Rotate to the right/left by $\Delta \theta$ degrees. (3) Tilt the camera upward/downward by $\Delta \varphi$ degrees. The environment provides textual feedback on action validity, such as collision detection. Additionally, we offer a script for automatic task generation, allowing users to create custom task datasets by specifying the configuration.

**EB-Manipulation.** EB-Manipulation extends VLMBench (Zheng et al., 2022) to evaluate MLLM-based embodied agents in low-level object manipulation. The agent controls a robotic arm using a 7-dimensional action vector, specifying movement parameters. Direct low-level manipulation is challenging for MLLMs. To overcome this challenge, we implemented enhancements, as illustrated in Figure 2: (1) action space discretization (Yin et al., 2024), which divides the position components $(x, y, z)$ into 100 bins and the orientation components $(roll, pitch, yaw)$ into 120 bins, enabling valid actions to take forms like $[x, y, z, roll, pitch, yaw, gripper] =$

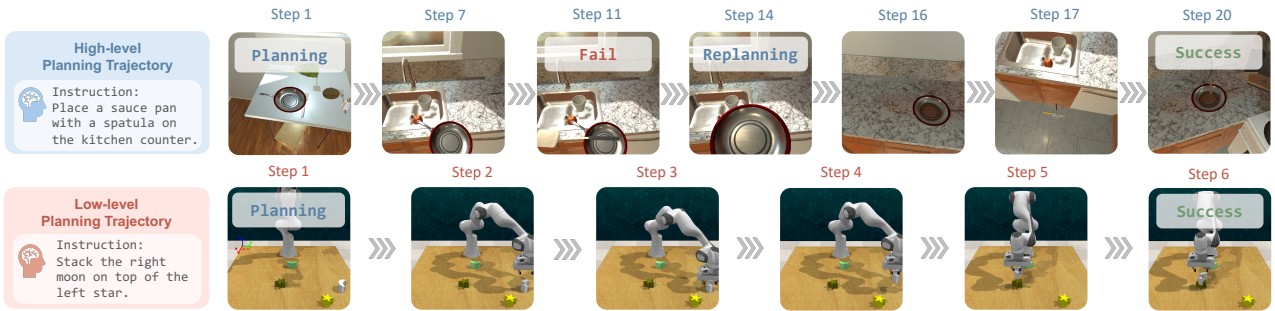

*Figure 3.* Planning examples in EB-ALFRED and EB-Manipulation based on GPT-4o.

[57, 61, 20, 10, 60, 25, 1]; and (2) additional information like YOLO (Redmon, 2016) detection boxes with index markers (Yang et al., 2023a) and object pose estimation for indexed objects, reducing the need for precise 3D location.

### 4.2. Capability-oriented Data Collection

We aim to collect capability-oriented data for our four environments. To accomplish this, we have identified six capability categories, as outlined in Table 5: (1) The **Base** subset evaluates basic task-solving skills necessary for planning action sequences across tasks of low to medium difficulty. (2) The **Common Sense** subset focuses on the use of common sense knowledge to indirectly refer to objects, such as describing a refrigerator as "a receptacle that can keep food fresh for several days." This subset evaluates the ability of embodied agents to reason using common sense. (3) The **Complex Instruction** subset includes relatively longer contexts, which can be relevant or irrelevant, to obscure the instruction. This measures an agent's ability to discern user intent from a long context. (4) The **Spatial Awareness** subset refers to objects by their location relative to other objects. (5) The **Visual Appearance** subset involves referring to objects based on their visual attributes, such as color or shape. (6) The **Long Horizon** subset comprises tasks requiring extended action sequences, typically more than 15 steps in EB-ALFRED. These subsets cover a broad range of scenarios, enabling a fine-grained evaluation of embodied agents' capabilities.

To construct a diverse dataset, we employ different data collection strategies. For EB-ALFRED and EB-Manipulation, data was gathered through a combination of manual annotation and instruction augmentation using GPT-4o (OpenAI, 2024a). For EB-Habitat, we reorganized and adapted an existing dataset from (Szot et al., 2023), aligning it with our specific objectives. Differently, data for EB-Navigation was generated entirely through automated Python programs. In summary, EB-ALFRED and EB-Habitat each include 300 test instances, with 50 instances for 6 subsets. Due to design challenges, EB-Navigation omits the spatial awareness subset and EB-Manipulation excludes the long-horizon subset.

EB-Navigation consists of 300 test cases distributed across 5 subsets (60 instances each), while EB-Manipulation contains a total of 228 instances, with 48 instances for each subset except visual appearance, which includes 36 instances. Detailed data collection is provided in Appendix C.

### 4.3. Vision-driven Agent Design

To evaluate MLLMs as agents in EMBODIEDBENCH, we design a unified embodied agent pipeline, illustrated in Figure 2. This pipeline provides a robust framework for processing multimodal inputs, reasoning through interactions, and generating structured, executable plans composed of sequential actions. Two planning examples are provided in Figure 3, with additional examples available in Appendix J. Below, we outline the key components of our agent design.

**Agent Input:** The agent processes a variety of inputs, including language instructions, visual perceptions, in-context demonstrations, interaction history, and task-specific information. For visual perception, the agent can utilize either the current step image or a sequence of historical images within a sliding window. *However, we observe that current MLLMs struggle to understand multiple historical images effectively, so we primarily rely on the current step image for efficiency.* An exception is made for EB-Navigation, which is discussed in more detail in Appendix G. Task-specific information varies by task type. For high-level tasks and EB-Navigation, the agent requires valid skill sets, while EB-Manipulation includes descriptions of the action format. Additionally, EB-Manipulation incorporates detection boxes with visual markers and object positions to help MLLMs accurately identify 3D locations. More examples of input prompts are provided in Appendix I.

**Task Planner:** At each planning step, the agent: (1) generates a textual description of the current visual input; (2) reflects on past actions and environmental feedback; (3) reasons about how to achieve the goal using available information; (4) formulates a language-based plan; and (5) converts it into an executable plan in the required format. All outputs are structured in JSON. Unlike prior work planning one action per timestep (Liu et al., 2024e), we support

multi-step planning, allowing the agent to dynamically decide the number of actions needed. It offers two advantages: (1) better alignment with in-context examples for sequential decision-making, and (2) reduced plan redundancy, especially in low-level tasks where single action causes limited changes in images, thereby minimizing MLLM API calls. If a plan fails or triggers an invalid action, the agent restarts planning from the latest state.

## 5. Experiments

In this section, we conduct comprehensive experiments to evaluate the performance of various MLLMs in EMBODIED-BENCH, followed by ablation studies in Sections 5.3 and 5.4 and error analysis in Section 5.5.

### 5.1. Experimental Setups

We benchmark 24 models, including 8 leading proprietary models and 16 SOTA open-source models. The proprietary models include GPT-4o and GPT-4o-mini (OpenAI, 2024a;b), Claude-3.5-Sonnet and Claude-3.7-Sonnet (Anthropic, 2024), Gemini Pro and Gemini Flash (Team et al., 2024a; DeepMind, 2024), and Qwen-VL-Max (Bai et al., 2023). The open-source models include InternVL2.5 and InternVL3 (8B / 38B / 78B) (Chen et al., 2025; Zhu et al., 2025), Qwen2-VL and Qwen2.5-VL (7B / 72B) (Wang et al., 2024; Bai et al., 2025), Gemma-3 (12B / 27B) (Team et al., 2025), Ovis2 (16B / 34B) (Lu et al., 2024), and LLaMA3.2 Vision Instruct (11B / 90B) (Meta, 2024). For consistency, all models are set with a temperature of 0 and a maximum completion token length of 2048. All images are standardized to a resolution of $500\times500$ pixels. The maximum number of environment steps is 30 for high-level tasks, 20 for EB-Navigation, and 15 for EB-Manipulation. We use the task success rate as the primary metric in our experiments. More results and ablations are deferred to Appendix F.

### 5.2. Benchmark Results

**Overall Results.** Tables 2 and 3 summarize the results for high-level and low-level tasks, respectively. Overall, *current MLLMs demonstrate strong performance on high-level tasks but struggle with low-level tasks, especially EB-Manipulation.* Among **proprietary models**, we observe that different models excel at different task levels: Claude-3.5-Sonnet achieves the highest average accuracy on high-level tasks, with 64.0% on EB-ALFRED and 68.0% on EB-Habitat, while GPT-4o leads in low-level tasks, scoring 57.7% on EB-Navigation and 28.9% on EB-Manipulation. For **open-source models**, InternVL3-78B delivers the strongest overall performance, surpassing several proprietary models and closely matching GPT-4o on low-level tasks with 53.7% on EB-Navigation and 26.3% on EB-Manipulation. Additionally, open-source models exhibit a clear scaling trend, with performance improving as model size increases. Nevertheless, a substantial performance gap remains between the top proprietary and open-source models, particularly on high-level tasks that demand advanced reasoning capabilities.

**The Role of Vision in Embodied Agent.** By comparing the performance of embodied agents with and without visual information (marked as "Lang") in Tables 2 and 3, we observe a clear distinction between low-level and high-level tasks. *Low-level tasks show a much stronger reliance on vision compared to high-level tasks.* For example, disabling vision causes GPT-4o's EB-Navigation performance to drop sharply from 57.7% to 17.4%, with long-horizon planning completely collapsing to 0%. This sharp decline highlights the critical importance of visual signals for low-level control tasks. Conversely, high-level tasks show much less dependence on visual input. GPT-4o (Lang) and GPT-4o-mini (Lang) perform on par with or even outperform their vision-enabled counterparts in EB-ALFRED and EB-Habitat, suggesting that these tasks may rely more heavily on textual information rather than visual input. We will further investigate the impact of language-centric factors in Section 5.3. These findings emphasize two key insights: (1) when designing MLLM-based embodied AI benchmarks, it is essential to consider action-level taxonomy, with greater attention to low-level action tasks, and (2) more advanced methods are needed to effectively leverage visual input for high-level embodied tasks.

**Fine-grained Results across Subsets.** We have the following findings based on our evaluation across 6 subsets.

*(1) Performance Varies across Different Subsets.* We observe that models perform differently across various subsets. For instance, while Claude-3.5-Sonnet is the best model on EB-Habitat overall, GPT-4o surpasses it on long-horizon subsets (64% vs. 58%). This divergence is even more evident in low-level tasks. In EB-Manipulation, for example, Claude-3.5-Sonnet scores 14.6 and 5.6 points higher than GPT-4o on the complex instruction and visual appearance subsets, respectively, but falls significantly behind on other capabilities. These results highlight the importance of fine-grained evaluations to uncover nuanced limitations in current models.

*(2) Long-Horizon Planning Is the Most Challenging Task.* The long-horizon subset consistently proves to be the most difficult, showing the largest performance gap compared to base scores. For instance, in EB-Habitat, Claude-3.5-Sonnet achieves 96% on the base subset but drops to 58% on the long-horizon subset. Similarly, GPT-4o falls from 86% to 64%. This trend holds true across both high-level and low-level tasks, suggesting that long-horizon planning remains a significant bottleneck for current MLLM-based agents.

*Table 2.* Task success rates on 6 subsets of EB-ALFRED and EB-Habitat, with the best proprietary model in bold and open-source model underlines per column. Success rates for subsets are integers since each subset consists of 50 test instances.

| Model | EB-ALFRED | | | | | | | EB-Habitat | | | | | | |
|---|---|---|---|---|---|---|---|---|---|---|---|---|---|---|
| | Avg | Base | Common | Complex | Visual | Spatial | Long | Avg | Base | Common | Complex | Visual | Spatial | Long |
| *Proprietary MLLMs* | | | | | | | | | | | | | | |
| GPT-4o | 56.3 | 64 | 54 | 68 | 46 | 52 | 54 | 59.0 | 86 | 44 | 56 | 68 | 36 | **64** |
| GPT-4o-mini | 24.0 | 34 | 28 | 36 | 24 | 22 | 0 | 32.7 | 74 | 22 | 32 | 22 | 32 | 14 |
| Claude-3.7-Sonnet | **67.7** | 68 | **68** | 70 | **68** | **62** | 70 | 58.7 | 90 | 58 | 58 | 62 | **38** | 46 |
| Claude-3.5-Sonnet | 64.0 | **72** | 66 | **76** | 60 | 58 | 52 | **68.0** | 96 | 68 | **78** | 70 | **38** | 58 |
| Gemini-1.5-Pro | 62.3 | 70 | 64 | 72 | 58 | 52 | 58 | 56.3 | 92 | 52 | 48 | 56 | **38** | 52 |
| Gemini-2.0-flash | 52.3 | 62 | 48 | 54 | 46 | 46 | 58 | 42.3 | 82 | 38 | 38 | 36 | 34 | 26 |
| Gemini-1.5-flash | 39.3 | 44 | 40 | 56 | 42 | 26 | 28 | 39.3 | 76 | 32 | 48 | 36 | 32 | 12 |
| Qwen-VL-Max | 41.3 | 44 | 48 | 44 | 42 | 38 | 32 | 45.3 | 74 | 40 | 50 | 42 | 30 | 36 |
| GPT-4o (Lang) | 58.0 | 62 | 64 | 70 | 52 | 46 | 54 | 56.0 | 82 | 52 | 58 | **74** | 34 | 36 |
| GPT-4o-mini (Lang) | 31.3 | 42 | 36 | 46 | 30 | 20 | 14 | 36.7 | 82 | 30 | 34 | 30 | 30 | 14 |
| *Open-Source MLLMs* | | | | | | | | | | | | | | |
| Llama-3.2-90B-Vision-Ins | 32.0 | 38 | 34 | 44 | 28 | 32 | 16 | 40.3 | 94 | 24 | 50 | 32 | 28 | 14 |
| Llama-3.2-11B-Vision-Ins | 13.7 | 24 | 8 | 16 | 22 | 6 | 6 | 25.0 | 70 | 16 | 28 | 10 | 20 | 6 |
| InternVL2_5-78B | 37.7 | 38 | 34 | 42 | 34 | 36 | 42 | 49.0 | 80 | 42 | 56 | 58 | 30 | 28 |
| InternVL2_5-38B | 23.3 | 36 | 30 | 36 | 22 | 14 | 26 | 38.3 | 60 | 28 | 48 | 34 | 32 | 28 |
| InternVL2_5-8B | 2.0 | 4 | 6 | 2 | 0 | 0 | 0 | 11.3 | 36 | 4 | 0 | 10 | 16 | 2 |
| InternVL3-78B | 39.0 | 38 | 34 | 46 | 42 | 38 | 36 | 55.0 | 84 | 58 | 60 | 56 | 32 | 40 |
| InternVL3-38B | 38.0 | 42 | 34 | 48 | 30 | 30 | 44 | 43.3 | 80 | 26 | 52 | 40 | 30 | 32 |
| InternVL3-8B | 10.3 | 20 | 14 | 14 | 12 | 0 | 2 | 24.3 | 60 | 14 | 24 | 18 | 20 | 10 |
| Qwen2-VL-72B-Ins | 33.7 | 40 | 30 | 40 | 30 | 32 | 30 | 35.7 | 70 | 30 | 36 | 32 | 28 | 18 |
| Qwen2-VL-7B-Ins | 1.7 | 6 | 0 | 2 | 0 | 0 | 2 | 18.3 | 48 | 6 | 16 | 20 | 18 | 2 |
| Qwen2.5-VL-72B-Ins | 39.7 | 50 | 42 | 42 | 36 | 34 | 34 | 37.7 | 74 | 28 | 42 | 40 | 24 | 18 |
| Qwen2.5-VL-7B-Ins | 4.7 | 10 | 8 | 6 | 2 | 0 | 2 | 14.3 | 32 | 2 | 26 | 10 | 14 | 2 |
| Ovis2-34B | 28.7 | 34 | 30 | 38 | 28 | 18 | 24 | 37.0 | 68 | 34 | 38 | 38 | 30 | 14 |
| Ovis2-16B | 16.3 | 26 | 16 | 24 | 12 | 16 | 4 | 32.0 | 66 | 26 | 42 | 28 | 22 | 8 |
| gemma-3-27b-it | 37.0 | 42 | 40 | 48 | 30 | 36 | 26 | 35.7 | 68 | 26 | 30 | 40 | 28 | 22 |
| gemma-3-12b-it | 25.7 | 32 | 26 | 38 | 26 | 20 | 12 | 23.0 | 58 | 10 | 24 | 18 | 24 | 4 |

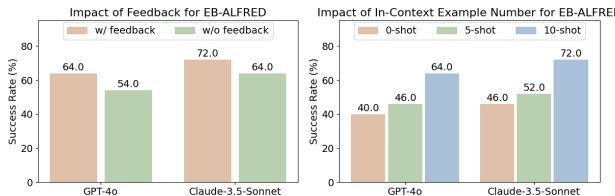

*Figure 4.* Language-centric ablations on EB-ALFRED.

## 5.3. Language-centric Ablation

We explore the role of the language-centric components, specifically focusing on **environment feedback** and **the number of in-context examples**. Comparisons are conducted using the base subset of EB-ALFRED. Our findings in Figure 4 reveal that removing environment feedback—which provides critical information during interaction—causes a 10% drop in success rate for GPT-4o and an 8% drop for Claude-3.5-Sonnet. Furthermore, while our experiments use 10 in-context examples by default, reducing this number significantly affects performance. In a 0-shot setting, the success rate drops to around 40%. When compared with results in Table 2, where removing vision can even lead to performance gains, these findings highlight that high-level tasks rely more heavily on textual information

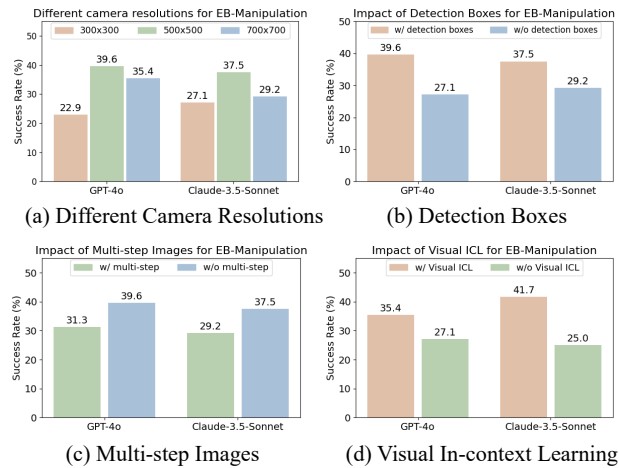

(a) Different Camera Resolutions  (b) Detection Boxes

(c) Multi-step Images  (d) Visual In-context Learning

*Figure 5.* Visual-centric ablations on EB-Manipulation.

than on visual input.

## 5.4. Visual-centric Ablation

Visual information is critical for the performance of low-level tasks. In this section, we thoroughly analyze the impact of four factors or potential enhancements: camera resolution, detection boxes, multi-step images, and visual in-context

*Table 3.* Task success rates on 5 subsets of EB-Navigation and EB-Manipulation, with the best proprietary model in bold and open-source model underlines per column.

| Model | EB-Navigation | | | | | | EB-Manipulation | | | | | |
|---|---|---|---|---|---|---|---|---|---|---|---|---|
| | Avg | Base | Common | Complex | Visual | Long | Avg | Base | Common | Complex | Visual | Spatial |
| *Proprietary MLLMs* | | | | | | | | | | | | |
| GPT-4o | **57.7** | 55.0 | 60.0 | **58.3** | **60.0** | **55.0** | **28.9** | **39.6** | **29.2** | 29.2 | 19.4 | 25.0 |
| GPT-4o-mini | 32.8 | 31.7 | 33.3 | 35.0 | 28.3 | 33.3 | 4.8 | 4.2 | 6.3 | 2.1 | 0.0 | 10.4 |
| Claude-3.7-Sonnet | 45.0 | 50.0 | 61.7 | 50.0 | 36.7 | 26.7 | 28.5 | 31.3 | 20.8 | **43.8** | **25.0** | 20.8 |
| Claude-3.5-Sonnet | 44.7 | **66.7** | 51.7 | 41.7 | 36.7 | 26.7 | 25.4 | 37.5 | 16.7 | 29.2 | 19.4 | 22.9 |
| Gemini-1.5-Pro | 24.3 | 23.3 | 25.0 | 25.0 | 28.3 | 20.0 | 21.1 | 14.6 | 14.6 | 22.9 | 16.7 | **35.4** |
| Gemini-2.0-flash | 48.7 | 63.3 | **65.0** | 50.0 | 51.7 | 13.3 | 16.7 | 14.6 | 8.3 | 14.6 | 13.9 | 31.3 |
| Gemini-1.5-flash | 41.7 | 56.7 | 50.0 | 46.7 | 50.0 | 5.0 | 9.6 | 14.6 | 10.4 | 4.2 | 8.3 | 10.4 |
| Qwen-VL-Max | 39.7 | 50.0 | 46.7 | 41.7 | 35.0 | 25.0 | 18.0 | 25.0 | 10.4 | 18.8 | 2.8 | 29.2 |
| GPT-4o (Lang) | 17.4 | 21.7 | 21.7 | 26.7 | 16.7 | 0.0 | 16.2 | 16.7 | 16.7 | 14.6 | 19.4 | 14.6 |
| GPT-4o-mini (Lang) | 8.3 | 3.3 | 13.3 | 10.0 | 15.0 | 0.0 | 6.6 | 12.5 | 0.0 | 2.1 | 2.8 | 14.6 |
| *Open-Source MLLMs* | | | | | | | | | | | | |
| Llama-3.2-90B-Vision-Ins | 30.0 | 48.3 | 23.3 | 38.3 | 33.3 | 6.7 | 14.9 | 10.4 | 12.5 | 16.7 | 10.4 | 20.8 |
| Llama-3.2-11B-Vision-Ins | 21.4 | 23.3 | 21.7 | 26.7 | 18.3 | 17.0 | 0.9 | 0.0 | 0.0 | 2.1 | 0.0 | 2.1 |
| InternVL2_5-78B | 30.7 | 36.7 | 38.3 | 33.3 | 21.7 | 23.3 | 18.0 | 16.7 | 16.7 | 14.6 | 22.2 | 20.8 |
| InternVL2_5-38B | 30.3 | 35.0 | 28.3 | 38.3 | 26.7 | 23.3 | 15.8 | 22.9 | 16.7 | 8.3 | 13.9 | 16.7 |
| InternVL2_5-8B | 21.3 | 35.0 | 23.3 | 21.7 | 26.7 | 0.0 | 7.0 | 8.3 | 2.1 | 6.3 | 8.3 | 10.4 |
| InternVL3-78B | 53.7 | 66.7 | 63.3 | 61.7 | 45.0 | 31.7 | 26.3 | 29.2 | 22.9 | 22.9 | 25.0 | 31.3 |
| InternVL3-38B | 50.7 | 55.0 | 61.7 | 55.0 | 56.7 | 25.0 | 22.6 | 20.8 | 14.6 | 20.8 | 19.4 | 37.5 |
| InternVL3-8B | 29.3 | 38.3 | 30.0 | 40.0 | 33.3 | 5.0 | 11.5 | 10.4 | 10.4 | 12.5 | 13.9 | 10.4 |
| Qwen2-VL-72B-Ins | 21.2 | 26.7 | 30.0 | 28.3 | 16.0 | 5.0 | 13.6 | 18.8 | 20.8 | 4.2 | 8.3 | 14.6 |
| Qwen2-VL-7B-Ins | 14.0 | 26.7 | 10.0 | 15.0 | 15.0 | 3.3 | 0.0 | 0.0 | 0.0 | 0.0 | 0.0 | 0.0 |
| Qwen2.5-VL-72B-Ins | 40.0 | 46.7 | 46.7 | 46.7 | 26.7 | 33.3 | 16.2 | 12.5 | 12.5 | 16.7 | 22.2 | 18.8 |
| Qwen2.5-VL-7B-Ins | 20.3 | 20.0 | 26.7 | 38.3 | 16.7 | 0.0 | 9.6 | 8.3 | 8.3 | 8.3 | 5.6 | 16.7 |
| Ovis2-34B | 45.7 | 63.3 | 50.0 | 56.7 | 46.7 | 11.7 | 26.8 | 31.3 | 25.0 | 18.8 | 27.8 | 31.3 |
| Ovis2-16B | 47.7 | 60.0 | 46.7 | 58.3 | 48.3 | 25.0 | 11.3 | 10.4 | 4.2 | 16.7 | 16.7 | 8.3 |
| gemma-3-27b-it | 45.4 | 53.3 | 45.0 | 61.7 | 50.0 | 16.7 | 17.5 | 25.0 | 16.7 | 16.7 | 8.3 | 20.8 |
| gemma-3-12b-it | 34.0 | 38.3 | 36.7 | 48.3 | 40.0 | 6.7 | 20.6 | 20.8 | 22.9 | 20.8 | 19.4 | 18.8 |

learning. All comparisons are based on the base subset of EB-Manipulation. Additional ablation results can be found in Appendix F.

**Camera Resolutions.** We investigate the effect of three camera resolutions on task performance. Our results, shown in Figure 5 (a), indicate that mid-range resolutions (500 × 500) achieve better results compared to both lower (300 × 300) and higher (700 × 700) resolutions. While low-resolution images may lack fine-grained details necessary for task execution, excessively high resolutions can introduce unnecessary complexity, making it harder for MLLMs to focus on relevant information for decision-making. These results highlight the importance of selecting an appropriate resolution when deploying MLLM-based embodied agents.

**Detection Boxes.** In EB-Manipulation, detection boxes and visual markers are used to align language instructions with visual information, helping to localize key objects in the scene. Figure 5 (b) shows that removing detection boxes reduces success rates from 39.6% to 27.1% for GPT-4o and from 37.5% to 29.2% for Claude-3.5-Sonnet, emphasizing

their important role in object localization for low-level tasks.

**Multi-step Image Input.** We also explore whether incorporating multi-step historical observations can enhance performance in our agent framework, as they may help address partial observability. For EB-Manipulation, we include observations from the past two steps in addition to the current step. Two multi-step image examples are shown in Figure 10 and 11. Figure 5 (c) presents the quantitative results. Our experiments reveal that current MLLMs struggle to effectively utilize multiple image inputs, often leading to confusion about their current state. Future work could focus on developing methods to better leverage multiple images for enhanced understanding and reasoning.

**Visual In-context Learning (ICL).** Previous work has primarily relied on text-based ICL demonstrations. In this study, we investigate the impact of visual ICL for embodied agents by including image observations as part of the in-context examples for EB-Manipulation. This approach helps the model better understand the relationship between successful low-level actions and the object positions in the

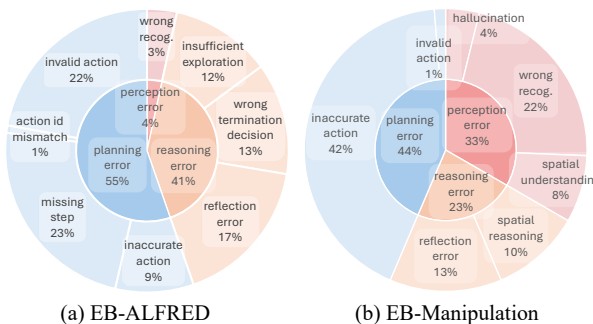

(a) EB-ALFRED      (b) EB-Manipulation

*Figure 6.* Error Analysis.

image. Visual ICL examples are demonstrated in Figure 15. We limit the number of examples to two to avoid overwhelming the model with excessive visual input. This may slightly lower the baseline performance, as the main results use more than two text-based examples. As shown in Figure 5 (d), the results demonstrate that visual ICL significantly outperforms language-only ICL. For instance, Claude-3.5-Sonnet achieves a 16.7% performance boost. These findings underscore the potential of visual ICL as a promising avenue for future research in embodied agents.

### 5.5. Error Analysis

We conducted an error analysis on GPT-4o to identify potential failure modes in EB-ALFRED and EB-Manipulation. For each environment, we sample 10 failure episodes from each subset, resulting in a total of 110 failed episodes to be analyzed. We found three main types of errors: perception errors, reasoning errors, and planning errors. Each error category corresponds to a specific stage in our agent pipeline, with definitions of sub-errors provided in Appendix H.

Overall, *planning errors are the most common issue in both environments, while perception errors are more prevalent in low-level tasks.* In EB-ALFRED, planning errors (55%) and reasoning errors (41%) dominate, while only 4% of errors are perception errors. Among planning errors, missing steps (23%) and invalid actions (22%) are the most common issues, highlighting challenges in generating complete and valid plans. Reflection errors (17%) suggest the model often fails to recognize planning mistakes in its action history. Another common failure is wrong termination errors (13%), where the model prematurely assumes the task is complete and stops too early. For EB-Manipulation, planning errors remain the primary cause of failure (44%), due to inaccurate actions, indicating difficulties in estimating precise gripper poses. Perception errors make up 33% of failures, with wrong recognition errors (22%) being the most frequent. These errors show that even with detection boxes annotated in the visual input, the model still fails to recognize object attributes correctly. This highlights considerable room for improvement in the visual capabilities of GPT-4o.

## 6. Conclusion

We introduce EMBODIEDBENCH, a comprehensive evaluation framework designed to assess MLLM-based embodied agents across tasks with varying action levels and capability-oriented subsets. Through extensive experiments, we identified key challenges, including difficulties in low-level manipulation and long-horizon planning, and the varying significance of vision input across tasks. By highlighting these areas for improvement, we hope EMBODIEDBENCH will inspire and guide future research toward building more capable and versatile vision-driven embodied agents.

## Limitations

A key limitation of this work is that our evaluation is conducted solely in simulated environments, without real-world experiments. This reflects a common trade-off between reproducibility, cost, safety, and real-world applicability. While real-world testing is essential for practical deployment, simulated benchmarks offer a standardized and reproducible setting, significantly reducing time, financial costs, and safety risks (Li et al., 2024c; Liu et al., 2024e). EMBODIEDBENCH represents a step forward in evaluating MLLM agents across diverse simulated embodied tasks. Future work could explore more realistic and complex simulations (Li et al., 2023) or develop standardized, cost-effective real-world test suites (Zhao et al., 2023; Fu et al., 2024) to bridge the gap toward practical deployment.

## Impact Statement

This work aims to advance the development of vision-driven embodied agents. There are many potential societal consequences of our work, none of which we feel must be specifically highlighted here. In addition, **a discussion of possible future directions is provided in Appendix B.**

## Acknowledgement

Tong Zhang acknowledges the support of NSF grant No. 2416897. Huan Zhang acknowledges the support of the University Research Program (URP) from Toyota Research Institute (TRI). This research reflects solely the opinions and conclusions of its authors, not those of TRI or any other Toyota entity. This research is also based upon work supported by U.S. DARPA ITM Program No. FA8650-23-C-7316 and DARPA ECOLE Program No. #HR00112390060. The views and conclusions contained herein are those of the authors and should not be interpreted as necessarily representing the official policies, either expressed or implied, of DARPA, or the U.S. Government. The U.S. Government is authorized to reproduce and distribute reprints for governmental purposes notwithstanding any copyright annotation therein.

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

# A. Additional Related Works

Foundation models (Bommasani et al., 2021), particularly Large Language Models (LLMs) (Brown et al., 2020; Achiam et al., 2023; Touvron et al., 2023; Yang et al., 2024a;c) and Multi-Modal Large Language Models (MLLMs) (Radford et al., 2021; Team et al., 2024a; Wang et al., 2024; Wu et al., 2024b; Du et al., 2025; Chen et al., 2024b; Xie et al., 2024), fundamentally transform how embodied agents perceive, make decisions, and act in physical and simulated environments.

The integration of these models into embodied agents evolves through several key approaches. Initially, Large Language Models (LLMs) are introduced to assist with high-level planning (Ahn et al., 2022; Huang et al., 2022a;b; Rana et al., 2023; Gao et al., 2024b; Huang et al., 2023d; Wang et al., 2023a; Huang et al., 2023b; Liu et al., 2023a; Wang et al., 2023b; Chen et al., 2023a; Huang et al., 2023a; Zhou et al., 2024a). They are also adopted for low-level controls (Mao et al., 2023; Yin et al., 2024). MLLMs are then incorporated for perception tasks such as object attribute identification, visual relation extraction, and action recognition (Xiao et al., 2022; Chen et al., 2023b; Wang et al., 2023d;e; Gao et al., 2024b; Gu et al., 2024). Subsequently, the role of MLLMs extends into policy-making through various approaches. Some works implement MLLMs in an end-to-end manner for direct action generation (Shridhar et al., 2022; Driess et al., 2023; Du et al., 2023; Yang et al., 2024d; Mu et al., 2024). Others enhance policy generation by using MLLMs to create visual markers or generate constraints or guidance with visual masks (Sharma et al., 2023; Stone et al., 2023; Nasiriany et al., 2024b; Huang et al., 2024a; Jiang et al., 2024). A different approach involves prompting MLLMs to generate code for creating policy or value functions (Liang et al., 2023; Huang et al., 2023c; 2024b).

Most recently, Vision Language Action Models (VLAs) (Brohan et al., 2022; 2023; Chi et al., 2023; Belkhale et al., 2024; Team et al., 2024b; Liu et al., 2024c; Kim et al., 2024) have emerged as a promising direction. These models typically utilize MLLMs or language-conditioned diffusion models as their foundation and are trained on low-level robotics action data. Another promising direction leverages world models as action simulators (Xiang et al., 2024; Agarwal et al., 2025; Liu et al., 2025). These approaches employ diffusion models conditioned on language inputs to predict future states given actions or task descriptions.

In response to the rapid advancements in this field, various simulators (Kolve et al., 2017; Puig et al., 2018; Shridhar et al., 2020a; Xiang et al., 2020; Shen et al., 2021; Li et al., 2021; 2023; Nasiriany et al., 2024a) and evaluation benchmarks (Shridhar et al., 2020b;a; Zheng et al., 2022; Li et al., 2023; Szot et al., 2023; Luo et al., 2023; Li et al., 2024a; Koh et al., 2024; Choi et al., 2024; Khanna et al., 2024; Liu et al., 2024e; Li et al., 2024b; Zhang et al., 2024a; Song et al., 2024) have been developed. However, existing benchmarks exhibit notable limitations. For instance, ALFWorld (Shridhar et al., 2020b), AgentBench (Liu et al., 2023b), Lota-bench (Choi et al., 2024), and Embodied Agent Interface (Li et al., 2024b) lack support for multimodal input evaluation. Furthermore, most benchmarks are narrowly focused on specific domains, particularly high-level household tasks (Shridhar et al., 2020a; Li et al., 2023; Szot et al., 2023), while others, such as VLMbench (Zheng et al., 2022) and GOAT-bench (Khanna et al., 2024), concentrate on low-level control for manipulation and navigation, respectively. Although VisualAgentBench (Liu et al., 2024e) pioneers the evaluation of MLLMs across multiple domains, it is limited to high-level tasks like household and Minecraft, and does not support fine-grained capability assessment. Embodied Agent Interface (Li et al., 2024b) and VLABench (Zhang et al., 2024a) introduce fine-grained evaluation metrics with language model support, but their focus remains primarily on LLMs and VLAs rather than MLLMs. Concurrently, EmbodiedEval (Cheng et al., 2025) proposes a multi-domain benchmark for evaluating MLLMs across navigation, object interaction, social interaction, attribute question answering, and spatial question answering. While it overlaps with our work in navigation and object interaction, it does not include low-level manipulation tasks or capability-oriented evaluation. Moreover, the benchmark is limited in scale, containing only 328 test instances.

# B. Future Research Directions

While EMBODIEDBENCH represents a significant step forward in evaluating MLLM-based embodied agents, several challenges remain, offering rich opportunities for future research. Below, we outline potential research directions:

- *Expanding Task Diversity.* Current benchmarks for MLLM-based embodied agents are still limited in task diversity. Future research could explore more realistic and complex environments with different action levels, such as autonomous driving (Gulino et al., 2024; Ma et al., 2024a; Gao et al., 2024a), multi-agent collaboration (Liu et al., 2024d), and human-agent interaction (Chang et al., 2024). These scenarios would better assess the agents' adaptability and generalization capabilities in real-world settings.

- **Low-Level Tasks and Spatial Reasoning.** Our findings show that current MLLM-based agents struggle with spatial reasoning and low-level control. Future research could improve these capabilities by better integrating spatial reasoning with low-level action planning, including 3D visual grounding (Chen et al., 2024a; Cheng et al., 2024) and alignment (Ahn et al., 2022; Yang et al., 2024d).

- **Long-Horizon Planning.** Long-horizon planning is still challenging for embodied agents. Future research can study techniques like hierarchical planning (Song et al., 2023; Ajay et al., 2023), memory-augmented methods (Sarch et al., 2024a), and world models (Mazzaglia et al., 2024) to enhance their ability to plan and execute complex, multi-step tasks more effectively.

- **Multi-step/Multi-view Image Understanding.** Our experiments show that current MLLMs struggle with multi-step and multi-view image inputs. Future research could improve multi-frame and multi-view comprehension, temporal reasoning, and spatial awareness to enhance MLLM agents' visual perception and reasoning. One promising direction is leveraging video pretraining (Madan et al., 2024; Wang et al., 2024) to better equip embodied agents for these challenges.

- **Visual In-context Learning (ICL).** Our experiments confirm the effectiveness of visual ICL (Zhou et al., 2024b; Sarch et al., 2024b) in embodied decision-making. This approach is promising because it enables adaptability and versatility without fine-tuning, allowing better use of off-the-shelf MLLMs. However, designing more effective visual ICL methods for embodied tasks remains an open problem for future research.

- **Training Multimodal Embodied Agents.** While our work focuses on evaluation, fine-tuning MLLMs for embodied tasks could significantly enhance their performance (Mu et al., 2024; Szot et al., 2024; Zawalski et al., 2024; Zhai et al., 2025). Future research can explore embodied pretraining, imitation learning, and both offline and online reinforcement learning (Sun, 2023) to better optimize MLLMs for embodied decision-making. Additionally, developing end-to-end learning approaches that seamlessly integrate perception, reasoning, and action could reduce the need for designing complex agent frameworks, leading to more adaptive and generalizable agents.

- **Robustness and Generalization of MLLM Agents.** Ensuring real-world applicability requires a thorough study of MLLM agents' robustness and generalization capabilities. While related studies are emerging in other domains (Zou et al., 2024; Xu et al., 2024; Yang et al., 2023b; 2024b; Zhang et al., 2024b), research on MLLM agents remains limited. Potential methods involve incorporating adversarial settings (Liu et al., 2024b; Wu et al., 2024a), dynamically generated environments (Wang et al., 2023c), or domain shifts (Chattopadhyay et al., 2021) to assess and enhance the ability of embodied agents to perform reliably in varying conditions.

By exploring these directions, the field can move closer to realizing the full potential of MLLM-based embodied agents in real-world applications.

## C. Details about EMBODIEDBENCH Environments and Datasets

Below, we provide detailed descriptions of four environments and their corresponding datasets. Please note that the maximum number of environment steps varies by task: 30 steps for high-level tasks (EB-ALFRED and EB-Navigation), 20 steps for EB-Navigation, and 15 steps for EB-Manipulation. In addition to task completion and exceeding the maximum step limit, we introduce two additional stopping conditions: (1) *Invalid Action Limit:* If the model generates more than 10 invalid actions in a single trajectory, indicating a lack of understanding and difficulty in producing valid actions. (2) *Empty Plan Generation:* If the model generates an empty plan because it incorrectly assumes the task is complete. This issue mainly occurs in high-level tasks, and once it happens, the model tends to keep generating empty plans without making progress. These additional stopping conditions help reduce unnecessary computational costs and improve evaluation efficiency.

### C.1. EB-ALFRED

**Task Description.** We develop the EB-ALFRED tasks based on the ALFRED dataset and the AI2-THOR simulator, which are well-regarded within the embodied AI community for their diverse household tasks and scenes. These tasks aim to evaluate an agent's ability to organize and execute sequences of high-level actions in household scenarios, such as "Put washed lettuce in the refrigerator." Each task in ALFRED can be described using the Planning Domain Definition Language (PDDL), which helps assess the agent's success in completing the task or subgoals. The ALFRED dataset includes 7 task

types, *Pick & Place, Stack & Place, Pick Two & Place, Clean & Place, Heat & Place, Cool& Place, and Examine in Light*. Our simulator is based on Lota-Bench's implementation for 8 high-level action types: "pick up", "open", "close", "turn on", "turn off", "slice", "put down", and "find". Each action can be parameterized with a specific object to form an action, e.g.,"find an apple" or "pick up an apple". The simulation offers an egocentric view and text feedback on the validity of action execution and potential reasons for any invalid actions. For example, it may indicate "failure to pick up an object because another object is already being held."

Despite its strengths, Lota-Bench's simulator has **three notable limitations**: (1) it does not support the *Pick Two & Place* task type due to the inability to handle multiple instances of one object type. (2) Some actions lead to incorrect task execution, such as the "put down" action erroneously placing an object on top of the sink instead of inside it, causing a correct action but unsuccessful outcome. (3) Additionally, some instructions in the original ALFRED dataset suffer from low quality. We observe the erroneous use of "potato" in task related to "tomato", which prevents agents from successfully completing the tasks due to these incorrect instructions.

To enhance the simulation, we implemented several improvements. Firstly, we introduced ***support for multi-instance settings*** in ALFRED by appending index suffixes to objects, such as "find a cabinet_2," to accommodate multiple instances of the same object type. Therefore, we can support all 7 task types in ALFRED. Given the dynamic number of objects in the ALFRED dataset, we made the action space of EB-ALFRED dynamic, ranging from 171 to 298 actions. To ***minimize redundancy in the action space***, we merge all "put down" actions into a single action, since only one object can be held at a time. Additionally, we manually ***corrected bugs in the original simulation and improved the quality of language instructions*** to ensure tasks are solvable and actions can be executed more accurately. These enhancements make EB-ALFRED a high-quality benchmark for evaluating embodied agents.

**Dataset Collection.** Following Lota-Bench (Choi et al., 2024), we use the valid seen set from the ALFRED dataset. We first partition the dataset based on the number of steps in the oracle policy. Specifically, we select 50 samples from the subset with fewer than 15 steps, carefully refining their instructions to minimize ambiguity and improve task solvability. The commonsense and complex instruction subsets are primarily derived from this base subset, with GPT-4o augmentation tailored to specific capabilities. Additionally, we select 50 tasks with more than 15 steps to form the long-horizon subset. The visual appearance and spatial awareness subsets are chosen directly from the original dataset based on language descriptions of color/shape, or relative positions. In total, EB-ALFRED comprises 300 testing instances, evenly distributed across six subsets (50 instances each).

## C.2. EB-Habitat

**Task Description.** EB-Habitat is developed based on the Language Rearrangement benchmark (Szot et al., 2023), featuring 282 diverse language instruction templates designed for robotic rearrangement tasks. It leverages the Habitat 2.0 simulator (Szot et al., 2021) and includes object data from the YCB dataset (Calli et al., 2015) and ReplicaCAD (Szot et al., 2021). The benchmark focuses on planning and executing 70 high-level skills to achieve user-defined goals, such as "Find a toy airplane and move it to the right counter." These skills are categorized into five action types: "navigation", "pick", "place", "open", and "close", each parameterized by specific objects.

Unlike ALFRED, which permits navigation to any object, EB-Habitat constrains navigation to receptacle-type objects, requiring robots to visit multiple locations to locate target items. Task and subgoal completion are evaluated using PDDL, with agents receiving visual input and textual feedback similar to ALFRED. Given its broad range of language instructions and distinct navigation constraints, EB-Habitat serves as a complementary counterpart to EB-ALFRED, expanding the scope of our high-level embodied tasks.

**Dataset Collection.** Habitat already provides fine-grained evaluation datasets with multiple subsets. We reorganize the subsets to formulate our dataset. Specifically, we merge "new scenes", "novel objects", and "instruction rephrasing" to form our base subset; we use the "context" set as our commonsense subset; we merge the "conditional instructions" and "irrelevant instruction text" as our complex instruction subset; we use the "referring expressions" as our visual appearance subset; we use the "spatial relationship" as our spatial awareness subset; we merge the "multiple rearrangements" and "multiple objects" as our long-horizon subset. Then, we sample 50 instances from each subset to form our EB-Habitat dataset, resulting in a total of 300 testing instances.

## C.3. EB-Navigation

**Task Description.** EB-Navigation is an evaluation suite built on AI2-THOR, designed to assess the navigation capabilities of embodied agents. In each task, the agent is placed at a starting position and must use visual observations and behavior feedback to execute low-level actions. The goal is to locate a target object and navigate to its vicinity. The agent's action space consists of seven actions that are executable by physical robots: (1) Move forward/backward by $\Delta x$. (2) Move rightward/leftward by $\Delta y$. (3) Rotate to the right/left by $\Delta \theta$ degrees. (4) Tilt the camera upward/downward by $\Delta \varphi$ degrees. At the start of each task, the agent is provided with a textual description of the action space, where each action is mapped to a unique index. Then, the agent selects an action by outputting the corresponding index, which the environment then executes.

At the beginning of each step, the environment provides the agent with a first-person visual observation. Using this visual input, the agent performs planning and decision-making to choose its next action. After executing an action, the environment evaluates its validity. For example, it checks for collisions or obstacles that might cause the action to fail. The environment then provides this valid or invalid signal as feedback to the agent. This signal is the only feedback the agent receives, as it is feasible to obtain in real-world scenarios. Together with the visual observations, this feedback equips the agent with sufficient information to perform navigation tasks effectively.

**Dataset Collection.** We constructed the dataset based on the original dataset provided by AI2-THOR. In AI2-THOR (Kolve et al., 2017), there are diverse scenes including environments such as kitchens, living rooms, and bedrooms, we designed a total of 90 navigation tasks, one for each scene. Each task dataset includes the following information: (1) *Initial Robot Pose*: Including its $(x, y, z)$ coordinates and initial orientation. (2) *Target object information*: Specifying the object type, ID and the 3D coordinates of the object's center. (4) *Language navigation instruction*: A human-readable instruction specifying the target object the agent needs to navigate to. We ensure the validness of the task dataset through the implementation of the following characteristics: (1) Initial distance: The agent's starting position is carefully constrained to be at least a certain adjustable distance (denoted as $\alpha$) from the target object. This adjustable $\alpha$ allows users to customize the number of navigation steps required for each task. (2) Target object accessibility: All target objects are exposed in the environment, reachable without requiring the agent to leave the ground. (3) Task completion conditions: A task is considered complete if the agent reaches a position within a specified distance threshold from the target object or if the maximum number of steps is exceeded. Additionally, the dataset includes an automated task-generation script. This script allows users to create custom task datasets by specifying parameters such as the target object type, initial distance threshold, and random seed for each scene. This flexibility ensures the dataset can be adapted to various research needs and scenarios.

For the capability-oriented subsets, we begin by sampling 60 instances from the original 90 tasks to form the base subset. We then use GPT-4 to perform instruction augmentation, generating more complex instructions and incorporating common sense knowledge to create the complex instruction and common sense subsets. The visual appearance subset is manually curated to include detailed descriptions of the target object's color and shape. Finally, the long horizon subset is constructed by ensuring the target object is not visible in the agent's initial view, requiring extended navigation to locate it. In total, we collect 300 testing instances across these 5 subsets (excluding the spatial awareness subset).

## C.4. EB-Manipulation

**Task description.** EB-Manipulation is an extension of VLMBench (Zheng et al., 2022) using the CoppeliaSim simulator (Rohmer et al., 2013) to control a 7-DoF Franka Emika Panda robotic arm. EB-Manipulation includes four task categories: (1) *Pick & Place Objects*, (2) *Stack Objects*, (3) *Shape Sorter Placement*, and (4) *Table Wiping*, each with randomly varied instances in color, position, shape, and orientation for diverse evaluation. The action space is a 7-dimensional vector. The simulator processes these actions and performs automatic motion planning to achieve the desired position. To facilitate motion planning, the environment operates in ABS_EE_POSE_PLAN_WORLD_FRAME mode, ensuring automatic trajectory execution from the current pose to the target pose. This simplifies the agent's role in predicting keypoints necessary for task completion.

Direct low-level manipulation is challenging for MLLMs due to insufficient domain-specific training. To overcome this, we implemented enhancements. **(1) Action space discretization**(Yin et al., 2024), which divides the position component into 100 bins and the orientation component into 120 bins, enabling valid actions to take forms like $[x, y, z, roll, pitch, yaw, gripper] = [57, 61, 20, 10, 60, 25, 1]$. Here, the first three dimensions (X, Y, Z) range from 0 to 100, while the next three (Pitch, Yaw, Gripper) range from 0 to 120. The gripper state remains binary (0.0 or 1.0). By discretizing the originally continuous action space, the model can predict actions using integer values, reducing complexity

for MLLMs. **(2) Additional information** like YOLO (Redmon, 2016) detection boxes with index markers and 3D object pose estimation for indexed objects, reducing the need for precise 3D location. Instead, the agent can focus on perceiving and reasoning about each object's relationship to the indexed objects. With these improvements and additional in-context examples, our MLLM agent effectively tackles complex low-level manipulation tasks.

At each step, the environment provides a front-view visual observation capturing a wooden table, a robotic arm positioned at the center corner, and multiple objects placed on the table. Each object is enclosed within a detection box labeled with a numerical index, sorted in descending order based on its Y-coordinate. A 3D XYZ coordinate system is displayed at the robot's frame origin for spatial reference. The field of view (FOV) and image resolution are configurable, offering flexibility in visual input settings. Additionally, all visible objects in the scene are provided with discrete 3D coordinates, sorted in descending order based on their Y-coordinate, and labeled with object index (e.g., "object 1"). This setup requires the agent to understand the correlation between objects mentioned in the instruction and their corresponding object indices. Using this position information, the agent can plan and execute action sequences to achieve the manipulation goal. To ensure validity, the environment evaluates each action, preventing constraint violations such as invalid trajectories or out-of-range movements. The validity signal serves as the sole feedback mechanism.

**Dataset Collection.**  For the base and spatial subsets, we select and curate samples from the VLMBench dataset. To generate instructions for each subset, we provide GPT-4o with 10 in-context examples. The common sense, visual appearance, and complex instruction subsets are derived from the base subset, with modifications designed to assess specific capabilities. The visual appearance subset consists of 36 tasks, as the table wiping task is excluded due to the inability to distinguish objects based on appearance. Each of the remaining 4 subsets comprises 48 tasks evenly distributed across four categories, with 12 tasks per category. In total, EB-Manipulation consists of 228 testing instances.

## D. Model Versions

Table 4 lists the versions or full names of the models used in our experiments. We accessed proprietary models through API calls and open-source models via local deployment using `lmdeploy` (Contributors, 2023) and `vllm` (Kwon et al., 2023).

## E. Definitions and Examples of Capability-oriented Subsets

As listed in Table 5, we provide definitions and examples of six capability-oriented subsets in EMBODIEDBENCH.

| Model Name | Creator | Full Name |
|---|---|---|
| GPT-4o | OpenAI | gpt-4o-2024-08-06 |
| GPT-4o-mini | OpenAI | gpt-4o-mini-2024-07-18 |
| Claude-3.7-Sonnet | Anthropic | claude-3-7-sonnet-20250219 |
| Claude-3.5-Sonnet | Anthropic | claude-3-5-sonnet-20241022 |
| Gemini-1.5-Pro | Google | gemini-1.5-pro |
| Gemini-2.0-flash | Google | gemini-2.0-flash-exp |
| Gemini-1.5-flash | Google | gemini-1.5-flash |
| Qwen-VL-Max | Qwen | qwen-vl-max-2025-01-25 |
| Llama-3.2-90B-Vision-Ins | Meta | meta-llama/Llama-3.2-90B-Vision-Instruct |
| Llama-3.2-11B-Vision-Ins | Meta | meta-llama/Llama-3.2-11B-Vision-Instruct |
| InternVL2_5-78B | OpenGVLab | OpenGVLab/InternVL2_5-78B |
| InternVL2_5-38B | OpenGVLab | OpenGVLab/InternVL2_5-38B |
| InternVL2_5-8B | OpenGVLab | OpenGVLab/InternVL2_5-8B |
| InternVL3-78B | OpenGVLab | OpenGVLab/InternVL3-78B |
| InternVL3-38B | OpenGVLab | OpenGVLab/InternVL3-38B |
| InternVL3-8B | OpenGVLab | OpenGVLab/InternVL3-8B |
| Qwen2-VL-72B-Ins | Qwen | Qwen/Qwen2-VL-72B-Instruct |
| Qwen2-VL-7B-Ins | Qwen | Qwen/Qwen2-VL-7B-Instruct |
| Qwen2.5-VL-72B-Ins | Qwen | Qwen/Qwen2.5-VL-72B-Instruct |
| Qwen2.5-VL-7B-Ins | Qwen | Qwen/Qwen2.5-VL-7B-Instruct |
| Ovis2-34B | AIDC-AI | AIDC-AI/Ovis2-34B |
| Ovis2-16B | AIDC-AI | AIDC-AI/Ovis2-16B |
| gemma-3-27b-it | Google | google/gemma-3-27b-it |
| gemma-3-12b-it | Google | google/gemma-3-12b-it |

*Table 4.* Full names of MLLMs used in our experiments.

*Table 5.* Definitions and examples of six capability-oriented subsets in EMBODIEDBENCH. Four environments EB-ALFRED, EB-Habitat, EB-Manipulation, and EB-Navigation are abbreviated as *ALF*, *Hab*, *Man*, and *Nav*, respectively.

| Subset Name | Instruction Example | Description |
|---|---|---|
| Base | *ALF:* Put washed lettuce in the refrigerator. 
 *Hab:* Move one of the pear items to the indicated sofa. 
 *Man:* Pick up the star and place it into the silver container. 
 *Nav:* Navigate to the pillow in the room and be as close as possible to it. | Instructions used to describe basic tasks. |
| Common Sense | *ALF:* Place washed leafy green vegetable in a receptacle that can keep it fresh for several days. 
 *Hab:* Prepare for a game by delivering something to play with to the TV stand. 
 *Man:* Pick up the bright object that usually appears in the night sky alongside the moon and place it into the silver box used for storing things. 
 *Nav:* I'm feeling thirsty and need a small container to hold water or coffee. Please navigate to that object and stay near it. | Refer to objects indirectly using common sense knowledge. |
| Complex Instruction | *ALF:* For freshness, place the washed lettuce in the refrigerator. This way, it's ready for any delightful recipe ideas you have. 
 *Hab:* When you find the fridge door open, go ahead and move an bowl to the sofa; otherwise, transport an hammer to the sofa. 
 *Man:* The objects on the desk seem perfect for children to play with. Can you now pick up the star and place it into the silver container? We're tidying up. 
 *Nav:* The rhythmic ticking of the kitchen clock blends with the occasional drip from the faucet. There's a small pile of onions on the table, freshly chopped. Please move towards the stove burner for me. The kitchen has a comforting hum to it. | Add longer relevant or irrelevant context to obscure the instruction. This is used to evaluate the ability of understanding complex instructions. |
| Spatial Awareness | *ALF:* Put two spray bottles in the cabinet under the sink against the wall. 
 *Hab:* Move a spatula from the right counter to the right receptacle of the left counter. 
 *Man:* Pick up the left object and place it into the front container. | Refer to objects by their location relative to other receptacles or objects. |
| Visual Appearance | *ALF:* Put a knife in a blue container onto the black table in the corner. 
 *Hab:* Deliver a small red object with green top to the intended a large gray piece of furniture with a backrest by physically moving it there. 
 *Man:* Put the green object with five evenly spaced points into the sorting container. 
 *Nav:* Find the rectangular yellowish object with a soft and smooth surface. | Refer to objects indirectly by their visual appearance. |
| Long Horizon | *ALF:* Pick up knife, slice apple, put knife in bowl, heat slice of apple in microwave, put apple slice on table. 
 *Hab:* Move the rubriks cube to the left counter, the wrench to the left counter, and the bowl to the brown table. 
 *Nav:* Navigate to the Toaster in the room and be as close as possible to it. | Describe a task that requires a long sequence of actions to complete. For EB-navigation, the instruction is similar to the base subset but the location of object is not visible at initialization. |

*Table 6.* **Subgoal success rates** on 6 subsets of EB-ALFRED and EB-Habitat, with the best proprietary model in bold and open-source model underlines per column.

| Model | EB-ALFRED | | | | | | | EB-Habitat | | | | | | |
|---|---|---|---|---|---|---|---|---|---|---|---|---|---|---|
| | Avg | Base | Common | Complex | Visual | Spatial | Long | Avg | Base | Common | Complex | Visual | Spatial | Long |
| *Proprietary MLLMs* | | | | | | | | | | | | | | |
| GPT-4o | 65.1 | 74.0 | 60.3 | 74.0 | 58.3 | **61.3** | 62.5 | 70.7 | 90.7 | 56.0 | 68.0 | 75.2 | **62.1** | **72.2** |
| GPT-4o-mini | 34.3 | 47.8 | 35.3 | 43.5 | 33.3 | 29.0 | 17.0 | 44.0 | 77.5 | 32.5 | 42.0 | 33.1 | 57.8 | 21.3 |
| Claude-3.5-Sonnet | 65.3 | 72.0 | 66.0 | 76.7 | **63.0** | 59.7 | 54.5 | **70.8** | **97.5** | **68.5** | **79.5** | 72.0 | 43.8 | 63.3 |
| Gemini-1.5-Pro | **67.4** | **74.3** | 66.7 | 76.5 | 62.8 | 59.0 | **65.0** | 61.0 | 92.5 | 53.5 | 49.5 | 59.4 | 50.0 | 61.2 |
| Gemini-2.0-flash | 56.3 | 65.7 | 51.3 | 58.3 | 50.7 | 50.0 | 62.0 | 48.2 | 82.0 | 39.5 | 43.0 | 39.0 | 49.6 | 36.2 |
| Gemini-1.5-flash | 46.1 | 49.5 | 45.2 | 60.2 | 48.3 | 32.2 | 41.5 | 46.8 | 79.0 | 33.0 | 50.0 | 41.2 | 55.5 | 22.0 |
| GPT-4o (Lang) | 65.6 | 67.7 | **70.3** | **77.0** | 59.7 | 54.0 | **65.0** | 66.7 | 85.2 | 58.5 | 67.5 | **79.2** | 62.1 | 47.7 |
| GPT-4o-mini (Lang) | 40.1 | 44.8 | 41.2 | 54.2 | 36.0 | 24.7 | 39.5 | 48.1 | 85.8 | 39.0 | 43.5 | 39.0 | 56.8 | 24.5 |
| *Open-Source MLLMs* | | | | | | | | | | | | | | |
| Llama-3.2-90B-Vision-Ins | 37.6 | 43.7 | 37.3 | 49.2 | 35.3 | 36.0 | 24.0 | 50.6 | 94.5 | 32.5 | 53.0 | 39.7 | 59.6 | 24.3 |
| Llama-3.2-11B-Vision-Ins | 19.7 | 29.7 | 13.0 | 25.7 | 28.7 | 9.3 | 12.0 | 33.2 | 72.0 | 23.8 | 36.5 | 16.2 | 39.7 | 11.2 |
| InternVL2_5-78B | 41.0 | 42.3 | 35.3 | 43.3 | 35.7 | 40.3 | 49.0 | 55.2 | 82.0 | 43.0 | 59.0 | 63.9 | 45.1 | 38.2 |
| InternVL2_5-38B | 31.3 | 37.3 | 33.0 | 38.3 | 25.3 | 17.3 | 36.5 | 44.0 | 61.5 | 32.5 | 49.0 | 39.5 | 46.3 | 35.0 |
| InternVL2_5-8B | 2.0 | 4.0 | 6.0 | 2.0 | 0.0 | 0.0 | 0.0 | 19.4 | 40.2 | 11.5 | 11.0 | 16.0 | 30.7 | 7.3 |
| Qwen2-VL-72B-Ins | 38.7 | 45.3 | 33.3 | 44.7 | 35.7 | 33.0 | 40.0 | 42.2 | 72.0 | 33.0 | 39.0 | 37.0 | 52.0 | 20.3 |
| Qwen2-VL-7B-Ins | 5.2 | 8.3 | 5.3 | 7.0 | 1.7 | 3.3 | 5.5 | 26.1 | 53.3 | 9.0 | 24.0 | 25.8 | 41.2 | 3.2 |

# F. Additional Experiment Results

To thoroughly evaluate the performance of MLLMs as agents within EMBODIEDBENCH, we present additional metric results, including subgoal success rate (Appendix F.1) and average step counts (Appendix F.2), and conduct a series of ablation studies. These ablation studies, spanning from Appendix F.3 to Appendix F.7, focus on five critical factors: (1) varying camera resolutions, (2) the use of detection boxes, (3) multi-step images, (4) multi-view images, and (5) visual in-context learning. In the subsequent sections, we systematically analyze each of these factors, offering insights into their effectiveness and potential limitations.

## F.1. Subgoal Success Rate

In addition to the task success rates presented in Table 2, we further analyze the subgoal success rates for high-level tasks (EB-ALFRED and EB-Habitat), as detailed in Table 6. Given the use of symbolic expressions (Planning Domain Definition Language, PDDL) in high-level tasks, calculating subgoal success rates is straightforward. For instance, a task success condition can be expressed as "condition A and condition B." Completing condition A alone results in a 50% subgoal success rate, even though the final task success rate remains 0%.

The results in Table 6 generally align with those in Table 2. For most models, the subgoal success rates are higher than their final task success rates, which is expected. Notably, Gemini-1.5-Pro achieves higher subgoal success rates than Claude-3.5-Sonnet on the EB-ALFRED benchmark, despite Gemini-1.5-Pro having a lower final task success rate. Additionally, GPT-4o demonstrates subgoal performance comparable to Claude-3.5, with a gap of less than 0.2 in both environments, despite a substantial gap in their final task success rates. These findings suggest that while models demonstrate better ability to achieve subgoals, completing the final task remains a significant challenge. Additionally, the capability to achieve subgoals may slightly differ from the ability to accomplish the entire task. Since our primary objective is to achieve the full task, future research should focus on developing strategies to improve the final task success rate.

## F.2. Average Planner and Environment Steps

This section presents the results of average planner steps and environment steps, which quantify the number of model inferences and interactions with the environment, respectively. Since we employ a multi-step planning strategy, the number of environment steps exceeds that of planner steps. However, it is important to note that neither planner steps nor environment steps serve as precise metrics for evaluating agent performance, unlike the success rate. This is because the agent may generate empty plans or produce more than 10 invalid actions, potentially triggering early termination. Consequently, fewer steps do not always indicate superior planning performance. Nevertheless, meaningful insights can still be derived from Table 7 and Table 8:

- The multi-step planning strategy demonstrates significant efficiency in most cases, reducing average planner steps by around 50% to 80% compared to average environment steps. This is particularly evident in the EB-Manipulation task, where the average planner step for GPT-4o is 2.6, and the average environment step is 12.9, resulting in nearly 80% fewer model inferences. This highlights the model's ability to generate long action sequences and we can effectively leverage this capability to minimize costs. Such efficiency is especially advantageous when utilizing expensive large proprietary MLLMs.

- Despite the inherent inaccuracies in average step counts, it is still possible to observe that more capable models tend to achieve smaller average planner and environment steps. For instance, Claude-3.5-Sonnet achieves the lowest planner and environment steps in both EB-ALFRED and EB-Habitat tasks, while GPT-4o records the lowest average planner and environment steps in EB-Manipulation. Additionally, larger models generally require fewer steps than their smaller counterparts, as evidenced by the comparison between GPT-4o and GPT-4o-mini, as well as Gemini-1.5-Pro and Gemini-2.0-Flash.

| Model | EB-ALFRED | | EB-Habitat | |
|---|---|---|---|---|
| | Avg Planner Step | Avg Env Steps | Avg Planner Step | Avg Env Steps |
| GPT-4o | 4.4 | 16.3 | 5.5 | 13.1 |
| GPT-4o-mini | 7.7 | 20.6 | 7.4 | 18.8 |
| Claude-3.5-Sonnet | 4.0 | 12.1 | 4.2 | 10.9 |
| Gemini-1.5-Pro | 3.9 | 15.7 | 5.4 | 12.6 |
| Gemini-2.0-flash | 4.4 | 16.3 | 6.8 | 14.8 |
| Llama-3.2-90B-Vision-Ins | 7.3 | 16.7 | 7.3 | 16.2 |
| InternVL2 5-78B | 5.5 | 13.9 | 6.3 | 14.1 |
| InternVL2_5-78B-MPO | 6.2 | 16.8 | 6.6 | 14.3 |
| Qwen2-VL-72B-Ins | 6.1 | 13.7 | 6.8 | 14.2 |

Table 7. Average planner steps and environment steps in EB-ALFRED and EB-Habitat for different models.

| Model | EB-Navigation | | EB-Manipulation | |
|---|---|---|---|---|
| | Avg Planner Step | Avg Env Steps | Avg Planner Step | Avg Env Steps |
| GPT-4o | 6.2 | 15.5 | 2.6 | 12.9 |
| GPT-4o-mini | 7.6 | 17.5 | 3.4 | 14.7 |
| Claude-3.5-Sonnet | 6.2 | 15.6 | 2.7 | 13.3 |
| Gemini-1.5-Pro | 8.8 | 16.5 | 2.7 | 13.4 |
| Gemini-2.0-flash | 9.2 | 16.0 | 2.8 | 14.0 |
| Llama-3.2-90B-Vision-Ins | 7.5 | 17.4 | 3.0 | 13.9 |
| InternVL2 5-78B | 13.2 | 17.3 | 2.9 | 13.5 |
| InternVL2_5-78B-MPO | 6.2 | 16.5 | 2.9 | 13.5 |
| Qwen2-VL-72B-Ins | 11.1 | 17.8 | 2.9 | 13.9 |

Table 8. Average planner steps and environment steps in EB-Navigation and EB-Manipulation for different models.

### F.3. Camera Resolution

As shown in Figure 7, we tested three camera resolutions—300×300, 500×500, and 700×700—on EB-ALFRED, EB-Manipulation, and EB-Navigation tasks. Our results reveal a task-dependent pattern: for EB-ALFRED, where vision plays a secondary role, increasing the resolution slightly improves performance for both GPT-4o and Claude-3.5-Sonnet, with

accuracy gains of $2\% \sim 4\%$. In contrast, for EB-Manipulation and EB-Navigation, resolution is more critical, with the best performance achieved at 500×500. This suggests that while low-resolution images may lack the fine details needed for task execution, overly high resolutions can introduce unnecessary complexity, making it harder for MLLMs to focus on relevant information. These findings underscore the importance of choosing the right resolution when deploying MLLM-based embodied agents.

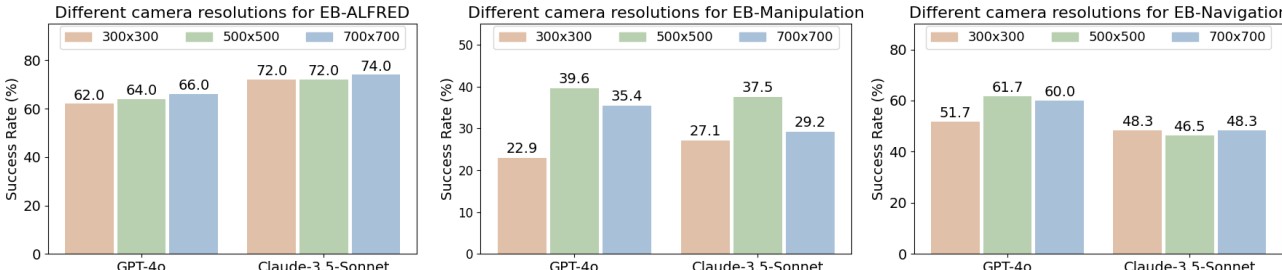

*Figure 7.* Impact of different camera resolutions on EMBODIEDBENCH.

## F.4. Detection Boxes

Figure 8 illustrates the impact of using detection (bounding) boxes. The results show that detection boxes are beneficial for both EB-ALFRED and EB-Manipulation, enhancing object recognition and interaction for models such as GPT-4o and Claude-3.5-Sonnet. In particular, EB-Manipulation experiences a significant performance boost of nearly $10\%$.

In contrast, detection boxes tend to hinder performance in EB-Navigation. This is likely because they obscure visual cues that are critical for effective path planning, leading to lower success rates. To investigate further, we conducted an additional experiment on EB-Navigation using only a single bounding box around the target object. As shown in Figure 9, this approach avoids the visual clutter caused by multiple boxes. The results, presented in Table 9, demonstrate that using a single detection box consistently improves accuracy. To better reflect real-world conditions, EB-Navigation omits detection boxes by default, requiring the MLLM agent to detect and recognize objects autonomously.

These findings highlight the importance of tailoring visual augmentation strategies to the specific requirements of each task. Consequently, we enable detection boxes by default only for EB-Manipulation, while disabling them for all other task groups.

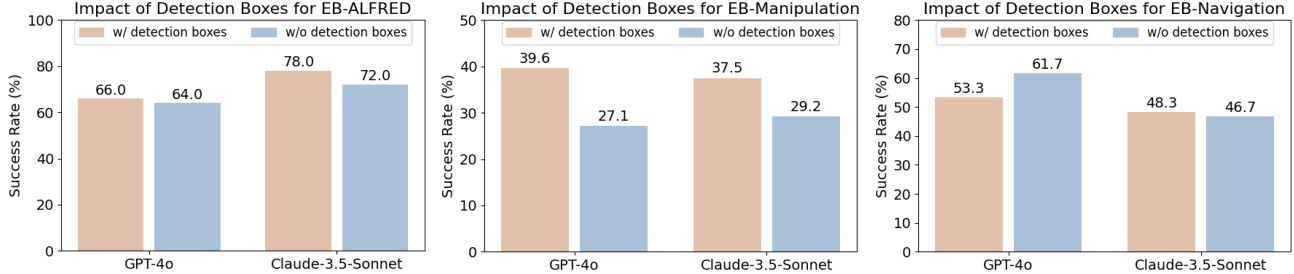

*Figure 8.* Impact of detection boxes on EMBODIEDBENCH.

*Table 9.* Comparing Different Detection Box Strategies in EB-Navigation.

| Model | No Box | One Box | Multi Box |
|---|---|---|---|
| GPT-4o | 61.7 | 68.3 | 53.3 |
| Claude-3.5-Sonnet | 46.7 | 58.3 | 48.3 |

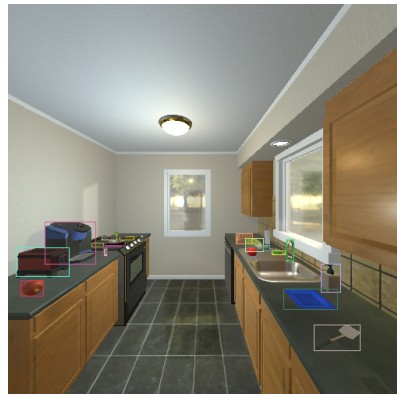 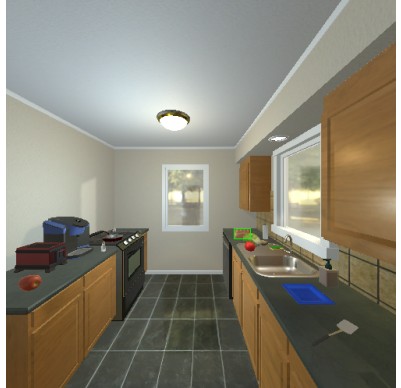

*Figure 9.* Illustration of Multiple (left) or Single (right) Detection Boxes in EB-Navigation

### F.5. Multi-step Images

Using sequences of images is a common approach to address partial observation. As shown in Figure 10 and 11, observation images from the previous two environment steps are also included in addition to the planner's original visual input. We explore the effectiveness of multi-step images—sequential frames shown in Figure 12—where the latest three images are included as input. Surprisingly, adding temporal context does not improve decision-making; instead, it leads to a decline in performance, particularly for EB-Manipulation. This may be due to the models' struggle to interpret the relationship between multiple sequential images and their current state. These results emphasize the challenges of effectively utilizing temporal continuity in vision-language tasks.

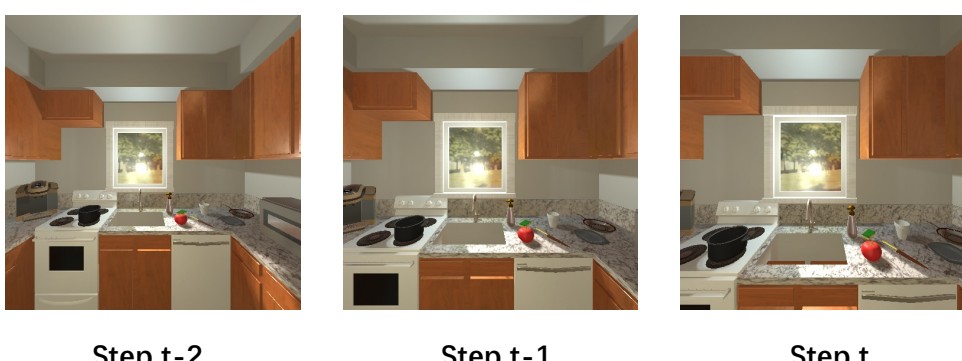

|  Step t-2  |  Step t-1  |  Step t  |

*Figure 10.* Multi-step observation example in EB-Navigation

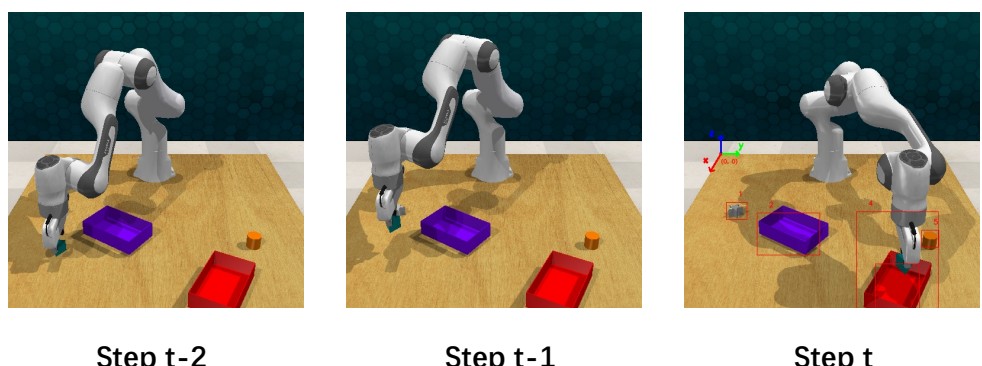

|  Step t-2  |  Step t-1  |  Step t  |

*Figure 11.* Multi-step observation example in EB-Manipulation

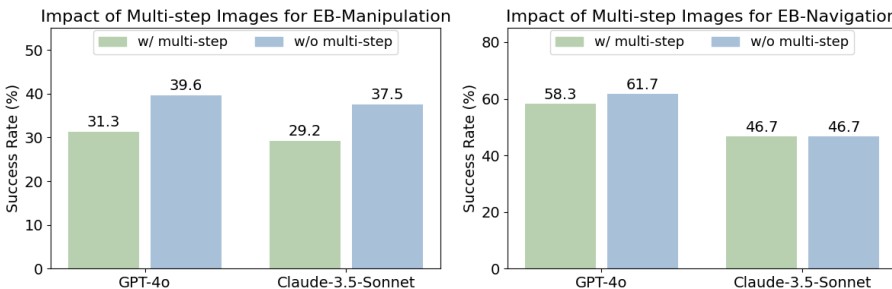

*Figure 12.* Impact of multi-step images on EMBODIEDBENCH.

### F.6. Multi-view Images

In addition to multi-step images, another approach is to incorporate multi-view images from different cameras at the same time step. As shown in Figure 13, the planner receives images from two different viewpoints as input. For EB-Navigation, the input consists of a front-view image and a top-down view image. For EB-Manipulation, the planner receives a front-view image and a wrist-view image. To evaluate whether multi-view images enhance performance in EB-Manipulation and EB-Navigation, we present the results in Figure 14. Surprisingly, using multi-view data also results in a performance decline, particularly for GPT-4o. While multiple viewpoints theoretically offer richer spatial context, GPT-4o and Claude-3.5-Sonnet seem to struggle with effectively integrating and leveraging these additional perspectives. This limitation may arise from challenges in multi-view feature fusion or the increased complexity of the input.

## EB-Navigation          EB-Manipulation

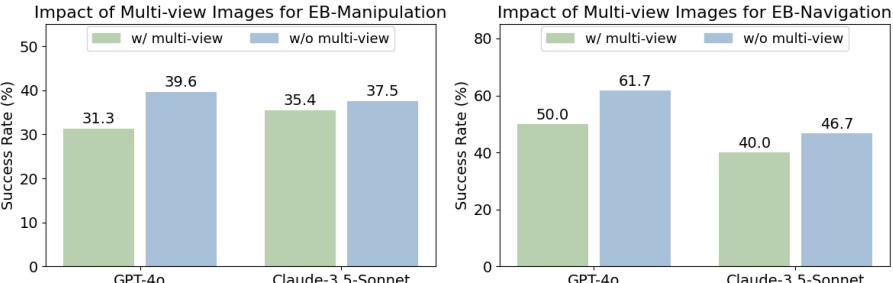

**Front view**   **Top-down view**          **Front view**   **Wrist view**

*Figure 13.* Multi-view observation example in EB-Navigation (left) and EB-Manipulation (right).

*Figure 14.* Impact of multi-view images on EMBODIEDBENCH.

## Visual in-context learning example for EB-Navigation

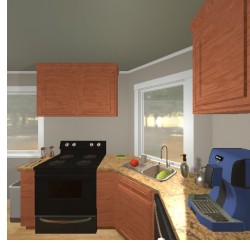

**Planning Step 1**

{"reasoning": "I can see the garbage can in the upper left corner of the image, next to the kitchen sink. To move there, we can go forward-left, but since there's a kitchen counter directly ahead, we should go left first. Following the strategy, I can go by first moving leftward.", "output": "[[3, 'Move leftward by 0.25'], [3, 'Move leftward by 0.25']]", "env_feedback": "Last action is executed successfully."}

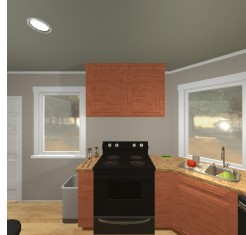

**Planning Step 2**

{"reasoning": "From the secene, I see that by moving leftward, we are getting closer to the garbage can. Now, the garbage can is in front of me, slightly to the left. And there's a large area ahead of us. Following the strategy, I can go by first moving forward then moving leftward.", "output": "[[0, 'Move forward by 0.25'], [0, 'Move forward by 0.25'], [0, 'Move forward by 0.25'],[0, 'Move leftward by 0.25']]", "env_feedback": "Last action is executed successfully."}

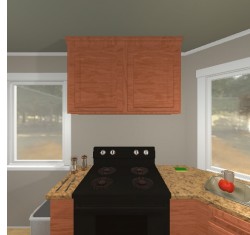

**Planning Step 3**

{"reasoning": "From the image we can see the garbage can is very close to us, still to our front-left. Moving leftward might be blocked but i can see that there is still space in front of me to get closer to the garbage can. Following the strategy, we can take about two steps forward then one step left to reach the garbage can.", "output": "[[3, 'Move forward by 0.25'],[3, 'Move forward by 0.25'], [3, 'Move leftward by 0.25']]", "env_feedback": "Success"}

## Visual in-context learning example for EB-Manipulation

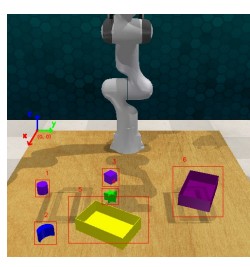

{"visual_state_description": "From left to right, I can see a purple cylinder at [45, 13, 18], a blue moon at [72, 20, 18], a green star at [50, 42, 17], a purple cube at [36, 42, 18], a yellow container at [70, 43, 17] and a purple container at [50, 77, 17].","reasoning_and_reflection": "The task is to pick up the star located at [50, 42, 17] and place it into the yellow container located at [70, 43, 17]. The plan involves moving the gripper to the star's position, closing the gripper to pick it up, moving to the yellow container's position, and opening the gripper to release the cube into the container.","executable plan": "[[50, 42, 17, 6, 61, 36, 1], [50, 42, 17, 6, 61, 36, 0], [47, 41, 26, 6, 61, 36, 0], [69, 39, 24, 88, 32, 77, 0], [69, 39, 24, 87, 32, 77, 1]]"}

*Figure 15.* Visual in-context learning examples for EB-Navigation & EB-Manipulation

### F.7. Visual In-context Learning (ICL)

Previous research has mainly focused on text-based in-context learning (ICL) demonstrations. In this study, we explore the impact of visual ICL for embodied agents by including image observations as part of the in-context examples for EB-Manipulation. This approach helps the model better grasp the connection between successful low-level actions and the positions of objects in the image. We provide two visual ICL examples in Figure 15, where the planner receives images corresponding to the textual in-context examples. To avoid overwhelming the model with excessive visual input, we limit the number of examples to two, which might slightly reduce performance compared to the main results without visual ICL. As illustrated in Figure 16, the results show that visual ICL significantly outperforms language-only ICL, with particularly

*Figure 16.* Impact of visual in-context learning on EMBODIEDBENCH.

impressive gains in manipulation tasks. For instance, Claude-3.5-Sonnet achieves a 16.7% improvement in performance. These findings highlight the potential of visual ICL as a promising direction for future research in vision-driven embodied agents.

### F.8. Additional Ablation Study Conclusion

Overall, our ablation studies reveal that while certain visual enhancements—such as moderate resolution increases, bounding-box detection, and visual in-context learning—can be beneficial, others—like extreme high-resolution inputs, multi-step/multi-view images, or detection boxes for navigation—may have limited or even negative effects. These findings highlight that the effectiveness of visual strategies heavily depends on the specific task and how additional visual information is integrated. Future research should focus on developing more advanced fusion techniques for embodied agents to better optimize the use of diverse visual inputs from multiple images.

## G. Further Discussion on Chat History as Input for EB-Navigation

Among the four environments in EMBODIEDBENCH, EB-Navigation is the most sensitive to historical input. Specifically, we observe that incorporating chat history—i.e., prior conversations paired with images, rather than the multi-step image setting studied in Appendix F.5, affects different models in varying ways.

Table 10 summarizes the results. We draw two key observations: (1) proprietary MLLMs tend to benefit from the inclusion of chat history, particularly in long-horizon tasks. (2) open-source models exhibit mixed preferences. For example, Llama-3.2 and the InternVL series perform better with chat history, whereas Qwen2-VL models perform best with single-step input, using only the summarized "interaction history" as in the other environments.

While we do not have a definitive explanation for this discrepancy, we hypothesize that some MLLMs may have been trained on large amount of multi-turn multimodal dialogue data, influencing their capability to this input format. Due to the lack of a clear conclusion, we adopt chat history only for EB-Navigation and do not use it as a global design choice.

## H. Error Definitions and Additional Analysis

### H.1. Error Type Definition

In this section, we define the types of errors and sub-errors encountered. We categorize errors into three main types: perception errors, reasoning errors, and planning errors. Each error type corresponds to a specific stage in our agent pipeline. For example, perception errors occur during the visual state description stage, reasoning errors arise in the reflection and reasoning stages, and planning errors occur during the language plan and executable plan generation stages. A detailed breakdown of sub-errors for each error type is provided in Table 11.

### H.2. Error Analysis for EB-Navigation

In evaluating the performance of MLLMs on navigation tasks, we identified three main types of errors: perception errors, reasoning errors, and planning errors. These errors significantly hinder the model's ability to successfully navigate to the target object.

*Table 10.* Comparison of performance in EB-Navigation with and without chat history. "w/o CH" indicates that the model does not use chat history as input.

| Model | EB-Navigation | | | | | |
|---|---|---|---|---|---|---|
| | **Avg** | **Base** | **Common** | **Complex** | **Visual** | **Long** |
| *Proprietary MLLMs* | | | | | | |
| GPT-4o | **57.7** | 55.0 | 60.0 | **58.3** | **60.0** | **55.0** |
| GPT-4o w/o CH | 45.3 | 63.3 | 58.3 | 51.7 | 38.3 | 15.0 |
| Claude-3.7-Sonnet | 45.0 | 50.0 | 61.7 | 50.0 | 36.7 | 26.7 |
| Claude-3.7-Sonnet w/o CH | 39.3 | 53.3 | 48.3 | 36.7 | 41.7 | 16.7 |
| Claude-3.5-Sonnet | 44.7 | **66.7** | 51.7 | 41.7 | 36.7 | 26.7 |
| Claude-3.5-Sonnet w/o CH | 36.3 | 45.0 | 40.0 | 38.3 | 46.7 | 11.7 |
| Gemini-1.5-Pro | 24.3 | 23.3 | 25.0 | 25.0 | 28.3 | 20.0 |
| Gemini-1.5-Pro w/o CH | 21.3 | 25.0 | 25.0 | 23.3 | 28.3 | 5.0 |
| Gemini-2.0-flash | 48.7 | 63.3 | **65.0** | 50.0 | 51.7 | 13.3 |
| Gemini-2.0-flash w/o CH | 36.0 | 48.3 | 35.0 | 43.3 | 43.3 | 10.0 |
| *Open-Source MLLMs* | | | | | | |
| Llama-3.2-90B-Vision-Ins | 30.0 | 48.3 | 23.3 | 38.3 | 33.3 | 6.7 |
| Llama-3.2-90B-Vision-Ins w/o CH | 22.3 | 35.0 | 31.7 | 18.3 | 18.3 | 8.3 |
| Llama-3.2-11B-Vision-Ins | 21.4 | 23.3 | 21.7 | 26.7 | 18.3 | 17.0 |
| Llama-3.2-11B-Vision-Ins w/o CH | 17.3 | 28.3 | 21.7 | 15.0 | 18.3 | 3.3 |
| InternVL2_5-78B | 30.7 | 36.7 | 38.3 | 33.3 | 21.7 | 23.3 |
| InternVL2_5-78B w/o CH | 29.3 | 40.0 | 28.3 | 35.0 | 26.7 | 16.7 |
| InternVL2_5-8B | 21.3 | 35.0 | 23.3 | 21.7 | 26.7 | 0.0 |
| InternVL2_5-8B w/o CH | 5.7 | 8.3 | 6.7 | 5.0 | 6.7 | 1.7 |
| Qwen2-VL-72B-Ins | 21.2 | 26.7 | 30.0 | 28.3 | 16.0 | 5.0 |
| Qwen2-VL-72B-Ins w/o CH | 36.3 | 36.7 | 40.0 | 50.0 | 36.7 | 18.3 |
| Qwen2-VL-7B-Ins | 14.0 | 26.7 | 10.0 | 15.0 | 15.0 | 3.3 |
| Qwen2-VL-7B-Ins w/o CH | 16.3 | 20.0 | 20.0 | 30.0 | 8.3 | 3.3 |

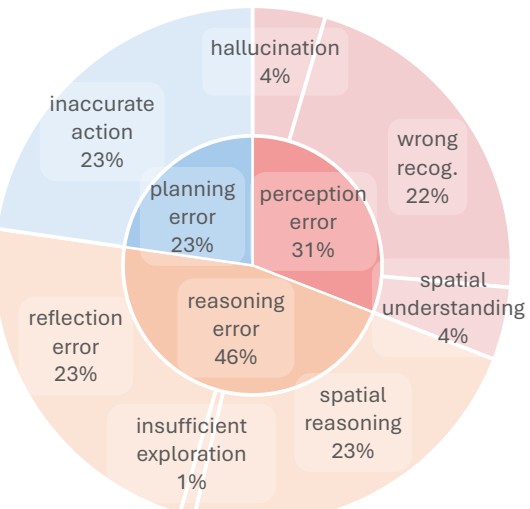

*Figure 17.* Error Analysis on EB-Navigation.

**Perception Errors.** The first category involves the model's ability to interpret visual observations and recognize the spatial position of the target object. We observed two common failure patterns: **(1) Wrong Recognition**: In some cases, the

| Error Type | Definition |
|---|---|
| **Perception Errors** | |
| *Hallucination* | Perceiving objects or attributes that are not present in the visual input |
| *Wrong Recognition* | Incorrectly identifying object types or attributes |
| *Spatial Understanding* | Misjudging object positions / depths in 3D space |
| **Reasoning Errors** | |
| *Spatial Reasoning* | Failure to understand / reason about spatial relationships |
| *Insufficient Exploration* | Only giving a suboptimal exploration strategy through reasoning |
| *Wrong Termination Decision* | Ending task execution before completing the goal |
| *Reflection Error* | Failing to realize previous errors or adapt plans using environmental feedback |
| **Planning Errors** | |
| *Inaccurate Action* | Executing actions with incorrect parameters / poses |
| *Missing Steps* | Omitting necessary actions in sequential plans |
| *Invalid Action* | Attempting physically impossible interactions |
| *Action ID Mismatch* | Misaligning action names with wrong action IDs |

*Table 11.* Error Taxonomy with Definitions

model failed to identify the target object even when it was present in the visual input. This suggests limitations in object recognition, possibly due to inadequate feature extraction or attention mechanisms. **(2)Hallucination of the Target Object**: In other instances, the model incorrectly claimed to have detected the target object when it was not actually present. This issue is particularly problematic, as it leads to premature conclusions and incorrect decisions instead of further exploration. Ideally, the model should acknowledge its inability to locate the target and continue navigating appropriately.

**Reasoning Errors.** The second category arises from the model's limitations in reasoning. This problem appears in two main ways: flawed logical reasoning about possible paths, even when visual observations are accurate; and weak reflection on spatial structure after failed attempts or feedback from previous steps. These issues indicate a lack of strong 3D spatial reasoning. The model often struggles to build a coherent 3D representation from sequential 2D observations, resulting in poor movement decisions. Furthermore, the model is not good at refining its actions based on ongoing feedback.

**Planning Errors.** Even when the model correctly identified the general direction, it often struggled with movement precision ("inaccurate actions"). For instance, it might overshoot the target by taking three steps instead of two. This highlights problems with spatial quantification, as the model's distance estimation and movement execution frequently did not align with real-world needs.

These main categories of errors reveal significant limitations in the navigation capabilities of current MLLM-based agents. To address these challenges, improvements are needed in several areas: strengthening object recognition, reducing hallucinations, enhancing 3D spatial reasoning, and aligning the model output with the action space to generate accurate plans. These advancements would enable more reliable and efficient autonomous navigation.

### H.3. Format Errors

In addition to the aforementioned error types, we also observe output format errors when generating JSON. Specifically, smaller-scale models can fail to produce valid JSON files. Table 12 shows the number of format errors across all six subsets, revealing that even proprietary models are not immune to these errors. A clear trend emerges: **larger models tend to have fewer format errors, while smaller models are more prone to such issues**. This highlights the need for further alignment of small models to improve the accuracy of structured outputs. Additionally, EB-ALFRED exhibits a higher number of errors, largely due to its greater complexity and the increased number of action steps in a trajectory.

| Model Name | EB-ALFRED | EB-Habitat |
|---|---|---|
| GPT-4o | 0.0067 | 0.0067 |
| GPT-4o-mini | 0.0867 | 0.0200 |
| Claude-3.5-Sonnet | 1.5400 | 0.0000 |
| Gemini-1.5-Pro | 0.0000 | 0.0000 |
| Gemini-2.0-flash | 0.0000 | 0.0000 |
| Llama-3.2-90B-Vision-Ins | 1.8033 | 0.0233 |
| Llama-3.2-11B-Vision-Ins | 1.6767 | 1.3233 |
| InternVL2 5-78B | 1.1467 | 0.0000 |
| InternVL2 5-38B | 2.4600 | 1.7433 |
| InternVL2 5-8B | 8.0400 | 4.3967 |
| Qwen2-VL-72B-Ins | 1.6100 | 0.0767 |
| Qwen2-VL-7B-Ins | 1.6933 | 0.7067 |

*Table 12.* Format error number per trajectory in EB-ALFRED and EB-Habitat across all subsets.

# I. Input of the Vision-driven Embodied Agent

## I.1. Prompts

We provide the agent input prompts used as textual input to MLLMs for all four environments.

---

Prompt for EB-ALFRED

## You are a robot operating in a home. Given a task, you must accomplish the task using a defined set of actions to achieve the desired outcome.
## Action Descriptions and Validity Rules
• Find: Parameterized by the name of the receptacle to navigate to. So long as the object is present in the scene, this skill is always valid.
• Pick up: Parameterized by the name of the object to pick. Only valid if the robot is close to the object, not holding another object, and the object is not inside a closed receptacle.
• Put down: Parameterized by the name of the object to put down to a nearby receptacle. Only valid if the robot is holding an object.
• Drop: Parameterized by the name of the object to put down. It is different from the Put down action, as this does not guarantee the held object will be put into a specified receptacle.
• Open: Parameterized by the name of the receptacle to open. Only valid if the receptacle is closed and the robot is close to the receptacle.
• Close: Parameterized by the name of the receptacle to close. Only valid if the receptacle is open and the robot is close to the receptacle.
• Turn on: Parameterized by the name of the object to turn on. Only valid if the object is turned off and the robot is close to the object.
• Turn off: Parameterized by the name of the object to turn off. Only valid if the object is turned on and the robot is close to the object.
• Slice: Parameterized by the name of the object to slice. Only valid if the object is sliceable and the robot is close to the object.
## The available action id (0 - {len(SKILL SET) - 1}) and action names are: {SKILL SET}.

{ICL EXAMPLES}

## Guidelines
1. **Output Plan**: Avoid generating empty plan. Each plan should include no more than 20 actions.
2. **Visibility**: Always locate a visible object by the 'find' action before interacting with it.
3. **Action Guidelines**: Make sure to match the action name and its corresponding action id in the output.
Avoid performing actions that do not meet the defined validity criteria. For instance, if you want to put an object in a receptacle, use 'put down' rather than 'drop' actions.
4. **Prevent Repeating Action Sequences**: Do not repeatedly execute the same action or sequence of actions. Try to modify the action sequence because previous actions do not lead to success.
5. **Multiple Instances**: There may be multiple instances of the same object, distinguished by an index following their names, e.g., Cabinet_2, Cabinet_3. You can explore these instances if you do not find the desired object in the current receptacle.
6. **Reflection on History and Feedback**: Use interaction history and feedback from the environment to refine and improve your current plan.
If the last action is invalid, reflect on the reason, such as not adhering to action rules or missing preliminary actions, and adjust your plan accordingly.

{ACTION HISTORY & ENVIRONMENT FEEDBACK (if available)}

## Now the human instruction is: {TASK INSTRUCTION} You are supposed to output in json. You need to describe the current visual state from the image, output your reasoning steps, and plan. At the end, output the action id (0 - {len(SKILL SET) - 1}) from the available actions to execute.

---

### Prompt for EB-Habitat

## You are a robot operating in a home. Given a task, you must accomplish the task using a defined set of actions to achieve the desired outcome.

## Action Descriptions and Validity Rules
• Navigation: Parameterized by the name of the receptacle to navigate to. So long as the receptacle is present in the scene, this skill is always valid.
• Pick: Parameterized by the name of the object to pick. Only valid if the robot is close to the object, not holding another object, and the object is not inside a closed receptacle.
• Place: Parameterized by the name of the receptacle to place the object on. Only valid if the robot is close to the receptacle and is holding an object.
• Open: Parameterized by the name of the receptacle to open. Only valid if the receptacle is closed and the robot is close to the receptacle.
• Close: Parameterized by the name of the receptacle to close. Only valid if the receptacle is open and the robot is close to the receptacle.

## The available action id (0 - 69) and action names are: {SKILL SET}.

{ICL EXAMPLES}

## Guidelines
1. **Output Plan**: Avoid generating empty plan. Each plan should include no more than 20 actions.
2. **Visibility**: If an object is not currently visible, use the "Navigation" action to locate it or its receptacle before attempting other operations.
3. **Action Validity**: Make sure to match the action name and its corresponding action id in the output.
Avoid performing actions that do not meet the defined validity criteria.
4. **Prevent Repeating Action Sequences**: Do not repeatedly execute the same action or sequence of actions.
Try to modify the action sequence because previous actions do not lead to success.
5. **Multiple Instances**: There may be multiple instances of the same object, distinguished by an index following their names, e.g., cabinet 2, cabinet 3. You can explore these instances if you do not find the desired object in the current receptacle.
6. **Reflection on History and Feedback**: Use interaction history and feedback from the environment to refine and enhance your current strategies and actions. If the last action is invalid, reflect on the reason, such as not adhering to action rules or missing preliminary actions, and adjust your plan accordingly.

{ACTION HISTORY & ENVIRONMENT FEEDBACK (if available)}

## Now the human instruction is: {TASK INSTRUCTION} You are supposed to output in json. You need to describe the current visual state from the image, output your reasoning steps, and plan. At the end, output the action id (0 - 69) from the available actions to execute.

> **Prompt for EB-Navigation at step 0**
>
> ## You are a robot operating in a home. You can do various tasks and output a sequence of actions to accomplish a given task with images of your status.
>
> ## The available action id (0 - 7) and action names are:
> action id 0: Move forward by 0.25,
> action id 1: Move backward by 0.25,
> action id 2: Move rightward by 0.25,
> action id 3: Move leftward by 0.25,
> action id 4: Rotate to the right by 90 degrees,
> action id 5: Rotate to the left by 90 degrees,
> action id 6: Tilt the camera upward by 30 degrees,
> action id 7: Tilt the camera downward by 30 degrees
>
> *** Strategy ***
>
> 1. Locate the Target Object Type: Clearly describe the spatial location of the target object from the observation image (i.e. on the front left side, a few steps from the current standing point).
>
> 2. Navigate by *** Using Move forward and Move right/left as the main strategy ***, since any point can be reached through a combination of those. When planning for movement, reason based on target object's location and obstacles around you.
>
> 3. Focus on the primary goal: Only address invalid action when it blocks you from moving closer in the direction to target object. In other words, do not overly focus on correcting invalid actions when direct movement toward the target object can still bring you closer.
>
> 4. *** Use Rotation Sparingly ***, only when you lose track of the target object and it's not in your view. If so, plan nothing but ONE ROTATION at a step until that object appears in your view. After the target object appears, start navigation and avoid using rotation until you lose sight of the target again.
>
> 5. *** Do not complete the task too early until you can not move any closer to the object, i.e. try to be as close as possible.
>
> {ICL EXAMPLES}
>
> ## Now the human instruction is: {TASK INSTRUCTION}. To achieve the task, 1. Reason about the current visual state and your final goal, and 2. Reflect on the effect of previous actions. 3. Summarize how you learned from the Strategy and Examples provided.
> Aim for about 2 actions in this step. !!!Notice: You cannot assess the situation until the whole plan in this planning step is finished and executed, so plan accordingly.
> At last, output the action id(s) (0 - 7) from the available actions to execute.
>
> The input given to you is a first-person view observation. Plan accordingly based on the visual observation.

Prompt for EB-Navigation at remaining steps

## You are a robot operating in a home. You can do various tasks and output a sequence of actions to accomplish a given task with images of your status.

## The available action id (0 - 7) and action names are:
action id 0: Move forward by 0.25,
action id 1: Move backward by 0.25,
action id 2: Move rightward by 0.25,
action id 3: Move leftward by 0.25,
action id 4: Rotate to the right by 90 degrees,
action id 5: Rotate to the left by 90 degrees,
action id 6: Tilt the camera upward by 30 degrees,
action id 7: Tilt the camera downward by 30 degrees

*** Strategy ***

1. Locate the Target Object Type: Clearly describe the spatial location of the target object from the observation image (i.e. on the front left side, a few steps from the current standing point).

2. Navigate by *** Using Move forward and Move right/left as the main strategy ***, since any point can be reached through a combination of those. When planning for movement, reason based on target object's location and obstacles around you.

3. Focus on the primary goal: Only address invalid action when it blocks you from moving closer in the direction to target object. In other words, do not overly focus on correcting invalid actions when direct movement toward the target object can still bring you closer.

4. *** Use Rotation Sparingly ***, only when you lose track of the target object and it's not in your view. If so, plan nothing but ONE ROTATION at a step until that object appears in your view. After the target object appears, start navigation and avoid using rotation until you lose sight of the target again.

5. *** Do not complete task too early until you can not move any closer to the object, i.e. try to be as close as possible.

{ICL EXAMPLES}

## Now the human instruction is: {TASK INSTRUCTION}.

{ACTION HISTORY & ENVIRONMENT FEEDBACK (if available)}

To achieve the task, 1. Reason about the current visual state and your final goal, and 2. Reflect on the effect of previous actions. 3. Summarize how you learned from the Strategy and Examples provided.
Aim for about 5-6 actions in this step to be closer to the target object. !!!Notice: You cannot assess the situation until the whole plan in this planning step is finished and executed, so plan accordingly.
At last, output the action id(s) (0 - 7) from the available actions to execute.

The input given to you is a first-person view observation. Plan accordingly based on the visual observation.

Prompt for EB-Manipulation

## You are a Franka Panda robot with a parallel gripper. You can perform various tasks and output a sequence of gripper actions to accomplish a given task with images of your status. The input space, output action space, and color space are defined as follows:

** Input Space **
- Each input object is represented as a 3D discrete position in the following format: [X, Y, Z].
- There is a red XYZ coordinate frame located in the top-left corner of the table. The X-Y plane is the table surface.
- The allowed range of X, Y, Z is [0, 100].
- Objects are ordered by Y in ascending order.

** Output Action Space **
- Each output action is represented as a 7D discrete gripper action in the following format: [X, Y, Z, Roll, Pitch, Yaw, Gripper].
- X, Y, Z are the 3D discrete positions of the gripper in the environment. It follows the same coordinate system as the input object coordinates.
- The allowed range of X, Y, Z is [0, 100].
- Roll, Pitch, and Yaw are the 3D discrete orientations of the gripper in the environment, represented as discrete Euler Angles.
- The allowed range of Roll, Pitch, and Yaw is [0, 120] and each unit represents 3 degrees.
- Gripper state is 0 for close and 1 for open.

** Color space **
- Each object can be described using one of the colors below:
["red", "maroon", "lime", "green", "blue", "navy", "yellow", "cyan", "magenta", "silver", "gray", "olive", "purple", "teal", "azure", "violet", "rose", "black", "white"],

Below are some examples to guide you in completing the task.

{ICL EXAMPLES}

## Now you are supposed to follow the above examples to generate a sequence of discrete gripper actions that completes the below human instruction.
Human Instruction: {TASK INSTRUCTION}
Input: {TASK INPUT}
Output gripper actions: {ACTION HISTORY & ENVIRONMENT FEEDBACK (if available)}

## I.2. Skill Sets

Below are the skill sets for EB-ALFRED and EB-Habitat. Note that the objects for EB-ALFRED vary depending on the scene, and the example provided here is illustrative. In contrast, the skill set for EB-Habitat remains static.

The skill sets (or action spaces) for EB-Navigation and EB-Manipulation are already included in the planner input prompt. For detailed prompts, please refer to Appendix I.1.

| Action Type | Target Object |
|---|---|
| Find | AlarmClock, Apple, Apple_2, Apple_3, ArmChair, BasketBall, Bathtub, Bed, Book, Box, Bread, Bread_2, ButterKnife, ButterKnife_2, Cabinet, Cabinet_10, Cabinet_2, Cabinet_3, Cabinet_4, Cabinet_5, Cabinet_6, Cabinet_7, Cabinet_8, Cabinet_9, Candle, Cart, CellPhone, CD, Chair, Cloth, CoffeeMachine, CoffeeTable, CounterTop, CounterTop_2, CounterTop_3, CreditCard, Cup, Cup_2, Cup_3, Desk, DeskLamp, DishSponge, DishSponge_2, DiningTable, Dresser, Drawer, Drawer_2, Drawer_3, Drawer_4, Drawer_5, Drawer_6, Egg, Faucet, FloorLamp, Fork, Fork_2, Fork_3, Fridge, GarbageCan, Glassbottle, HandTowel, Kettle, Kettle_2, Kettle_3, KeyChain, Knife, Knife_2, Ladle, Laptop, Lettuce, Lettuce_2, Microwave, Mug, Mug_2, Mug_3, Newspaper, Ottoman, Pan, Pan_2, Pan_3, PepperShaker, PepperShaker_2, PepperShaker_3, Pencil, Pen, Pillow, Plate, Plunger, Potato, Potato_2, RemoteControl, Safe, SaltShaker, SaltShaker_2, Shelf, SideTable, Sink, SoapBar, SoapBottle, SoapBottle_2, Sofa, Spatula, Spatula_2, SprayBottle, Statue, StoveBurner, StoveBurner_2, StoveBurner_3, StoveBurner_4, TennisRacket, TissueBox, Tomato, Toilet, ToiletPaper, ToiletPaperHanger, Vase, Watch, WateringCan, WineBottle |
| Pick up | AlarmClock, Apple, BaseballBat, BasketBall, Book, Bowl, Box, Bread, ButterKnife, Candle, CD, CellPhone, Cloth, CreditCard, Cup, DishSponge, Egg, Fork, Glassbottle, HandTowel, Kettle, KeyChain, Knife, Ladle, Laptop, Lettuce, Mug, Newspaper, Pan, Pen, Pencil, PepperShaker, Plate, Plunger, Potato, RemoteControl, SaltShaker, SoapBar, SoapBottle, Spatula, SprayBottle, Spoon, Statue, TennisRacket, TissueBox, Tomato, Vase, Watch, WateringCan, WineBottle |
| Put down | Object in hand |
| Drop | Object in hand |
| Open | Box, Cabinet, Cabinet_10, Cabinet_2, Cabinet_3, Cabinet_4, Cabinet_5, Cabinet_6, Cabinet_7, Cabinet_8, Cabinet_9, Drawer, Drawer_2, Drawer_3, Drawer_4, Drawer_5, Drawer_6, Fridge, Laptop, Microwave, Safe |
| Close | Box, Cabinet, Cabinet_10, Cabinet_2, Cabinet_3, Cabinet_4, Cabinet_5, Cabinet_6, Cabinet_7, Cabinet_8, Cabinet_9, Drawer, Drawer_2, Drawer_3, Drawer_4, Drawer_5, Drawer_6, Fridge, Laptop, Microwave, Safe |
| Turn on | DeskLamp, Faucet, FloorLamp, Microwave |
| Turn off | DeskLamp, Faucet, FloorLamp, Microwave |
| Slice | Apple, Bread, Lettuce, Potato, Tomato |

*Table 13.* The skill set for EB-ALFRED

| Action Type | Target Object |
|---|---|
| Navigate to | Cabinet 4, Cabinet 5, Cabinet 6, Cabinet 7, Chair 1, Left counter in the kitchen, Left drawer of the kitchen counter, Refrigerator, Refrigerator push point, Right counter in the kitchen, Right drawer of the kitchen counter, Sink in the kitchen, Sofa, Table 1, Table 2, TV stand |
| Pick up | Apple, Ball, Banana, Block, Book, Bowl, Box, Can, Clamp, Cleanser, Cup, Drill, Hammer, Knife, Lego, Lemon, Lid, Mug, Orange, Padlock, Peach, Pear, Plate, Plum, Rubik's cube, Scissors, Screwdriver, Spatula, Spoon, Sponge, Strawberry, Toy airplane, Wrench |
| Place at | Chair 1, Left counter in the kitchen, Left drawer of the kitchen counter, Refrigerator, Right counter in the kitchen, Right drawer of the kitchen counter, Sink in the kitchen, Sofa, Table 1, Table 2, TV stand |
| Open | Cabinet 4, Cabinet 5, Cabinet 6, Cabinet 7, Refrigerator |
| Close | Cabinet 4, Cabinet 5, Cabinet 6, Cabinet 7, Refrigerator |

*Table 14.* The skill set for EB-Habitat

## I.3. In-context examples

The in-context examples provided to the agent are detailed below. Each environment includes one to ten representative examples, with the complete set in our source code.

Specifically, **EB-ALFRED** utilizes 10 examples from the training set of the original ALFRED dataset for all subsets except the long-horizon subsets. We make sure the 10 examples diversely cover all 7 task types of ALFRED. For the long-horizon subsets, we include 7 different demonstrations closely related to the task types "pick_heat_then_place_in_recep" and "pick_coll_then_place_in_recep." It's important to note that the in-context examples do not include scenarios with multiple instances of the same object. Additionally, we ensure the examples do not involve variable action indices (i.e., for actions with IDs below 162, object names remain static without suffixes, while actions with IDs 162 or higher involve changing objects).

Similarly, **EB-Habitat** features 10 demonstrations from the 'train' set of the original Language Rearrangement dataset (Szot et al., 2023) across all subsets. For **EB-Navigation**, we provide 3 demonstrations for all subsets, ensuring they are distinct from the testing instances. In **EB-Manipulation**, the number of examples varies depending on task complexity: 2 examples for Pick & Place, 4 examples for Stacking and Shape Sorting, and 8 examples for Table Wiping. All demonstrations are carefully selected to differ from the testing instances.

In-context examples for EB-ALFRED tasks excluding long horizon tasks

## Example 1:
Human instruction: Pick up the alarm clock and turn on the lamp.
Output:

```
{
  "language_plan": "First, locate the alarm clock in the room. Once found, carefully
      pick up the alarm clock. Next, identify the location of the desk lamp. After
      locating the desk lamp, proceed to turn it on.",
  "executable_plan": [
    {"action_id": 57, "action_name": "find a AlarmClock" },
    {"action_id": 98, "action_name": "pick up the AlarmClock" },
    {"action_id": 63, "action_name": "find a DeskLamp" },
    {"action_id": 151, "action_name": "turn on the DeskLamp" }
  ]
}
```

## Example 2:
Human instruction: Examine a credit card by the light of a tall lamp.
Output:

```
{
  "language_plan": "Start by locating the credit card in the room. Once you have
      found the credit card, proceed to pick it up. Then, search for the floor lamp
      and find its location. After finding the floor lamp, turn it on to illuminate
      the area so you can examine the credit card effectively.",
  "executable_plan": [
    {"action_id": 56, "action_name": "find a CreditCard" },
    {"action_id": 96, "action_name": "pick up the CreditCard" },
    {"action_id": 27, "action_name": "find a FloorLamp" },
    {"action_id": 153, "action_name": "turn on the FloorLamp" }
  ]
}
```

## Example 3:
Human instruction: Use lamp light to look at the remote.
Output:

```
{
  "language_plan": "First, locate the remote control in the room. Once the remote
      control is found, pick it up carefully. Next, search for the floor lamp and
      determine its location. After you have located the floor lamp, turn it on to
      provide sufficient light for examining the remote control.",
  "executable_plan": [
    {"action_id": 68, "action_name": "find a RemoteControl" },
    {"action_id": 116, "action_name": "pick up the RemoteControl" },
    {"action_id": 27, "action_name": "find a FloorLamp" },
    {"action_id": 153, "action_name": "turn on the FloorLamp" }
  ]
}
```

In-context example for EB-ALFRED long horizon tasks

## Example 1:
Human instruction: Place the microwaved slice of apple on to the white table.
Output:

```
{
  "language_plan": "Locate an apple and pick it up, find a dining table and put the
      apple down, look for a knife and pick it up, find the apple and slice it.
      After slicing, I need to find a place like the dining table to put the knife
      down. After that, find an apple slice and pick it up, search for a microwave
      and open it, put the apple in the microwave and close it, turn on the
      microwave to heat the apple slice, turn off the microwave and open it, pick up
       the microwaved apple slice, close the microwave, find the white table, and
      place the microwaved apple slice on the white table.",
  "executable_plan": [
    {"action_id": 22, "action_name": "find a Apple" },
    {"action_id": 107, "action_name": "pick up the Apple" },
    {"action_id": 18, "action_name": "find a DiningTable" },
    {"action_id": 133, "action_name": "put down the object in hand" },
    {"action_id": 34, "action_name": "find a Knife" },
    {"action_id": 127, "action_name": "pick up the Knife" },
    {"action_id": 22, "action_name": "find a Apple" },
    {"action_id": 160, "action_name": "slice the Apple" },
    {"action_id": 18, "action_name": "find a DiningTable" },
    {"action_id": 133, "action_name": "put down the object in hand" },
    {"action_id": 22, "action_name": "find a Apple" },
    {"action_id": 107, "action_name": "pick up the Apple" },
    {"action_id": 38, "action_name": "find a Microwave" },
    {"action_id": 143, "action_name": "open the Microwave" },
    {"action_id": 133, "action_name": "put down the object in hand" },
    {"action_id": 144, "action_name": "close the Microwave" },
    {"action_id": 149, "action_name": "turn on the Microwave" },
    {"action_id": 150, "action_name": "turn off the Microwave" },
    {"action_id": 143, "action_name": "open the Microwave" },
    {"action_id": 22, "action_name": "find a Apple" },
    {"action_id": 107, "action_name": "pick up the Apple" },
    {"action_id": 144, "action_name": "close the Microwave" },
    {"action_id": 18, "action_name": "find a DiningTable" },
    {"action_id": 133, "action_name": "put down the object in hand" }
  ]
}
```

In-context examples for EB-Habitat

## Example 1:
Human instruction: Put both a toy airplane and a bowl onto the black table.
Output:

```
{
  "language_plan": "To achieve the goal, the robot must locate the toy airplane and
      the bowl in the room and then move each item to the black table. There are two
       tables in the room, and the robot identifies table 1 as the black table.
      Therefore, the objective is to place both objects on table 1. The plan is as
      follows: first, navigate to the sofa, pick up the airplane, move to table 1,
      and place the airplane there. Then, proceed to table 2, where the bowl might
      be, pick up the bowl, return to table 1, and set the bowl there.",
  "executable_plan": [
    {"action_id": 12, "action_name": "navigate to the sofa" },
    {"action_id": 47, "action_name": "pick up the toy airplane" },
    {"action_id": 6, "action_name": "navigate to table 1" },
    {"action_id": 50, "action_name": "place at table 1" },
    {"action_id": 7, "action_name": "navigate to table 2" },
    {"action_id": 42, "action_name": "pick up the bowl" },
    {"action_id": 6, "action_name": "navigate to table 1" },
    {"action_id": 50, "action_name": "place at table 1" }
  ]
}
```

## Example 2:
Human instruction: I made a mistake and left the fridge open. Can you assist me by closing it?
Output:

```
{
  "language_plan": "The objective is for the robot to close the refrigerator. To do
      so, the robot first navigates to the refrigerator and then closes it.",
  "executable_plan": [
    {"action_id": 13, "action_name": "navigate to the refrigerator" },
    {"action_id": 61, "action_name": "close the refrigerator" }
  ]
}
```

## Example 3:
Human instruction: Displace the sponge from the left counter.
Output:

```
{
  "language_plan": "To accomplish this task, the robot should navigate to the left
      counter to pick up the sponge. It should then move to another location, such
      as table 2, and place the sponge there.",
  "executable_plan": [
    {"action_id": 11, "action_name": "navigate to the left counter in the kitchen"},
    {"action_id": 28, "action_name": "pick up the sponge" },
    {"action_id": 7, "action_name": "navigate to table 2" },
    {"action_id": 51, "action_name": "place at table 2" }
  ]
}
```

In-context examples for EB-Navigation

## Example 1:
Human Instruction : Navigate to the floorlamp and be as close as possible.

Reasoning and reflection: I can see the floorlmap is at left side in front of me, next to the TVStand. Around me is a Cabinet at my left side. Following the guideline, since the floorlamp is at left side, I can approach it by a series of moving forward and left / rightward. Since currently there is an obstacle at my left, I will first move forward twice.
Executable plans: [[0, Move forward by 0.25], [0, Move forward by 0.25]]

Environment Feedback: Step 1 is executed successfully; Step 2 is executed successfully;

Reasoning and reflection: I can see now I get closer to the floorlamp. It is still located in front of me at the left. I will try move forward and move leftward in series.
Executable plans: [[0, Move forward by 0.25], [3, Move leftward by 0.25], [0, Move forward by 0.25], [3, Move leftward by 0.25], [0, Move forward by 0.25]]

Environment Feedback: Step 3 is executed successfully; Step 4 is executed successfully; Step 5 is executed successfully; Step 6 is executed successfully; Step 7 is Invalid, as a StandardIslandHeight is blocking you from moving forward.

Reasoning and reflection: I am a lot closer to the floorlamp by a series a forward and leftward movement. It is still in front of me. However, the last action is invalid, showing that I am blocked from moving forward. That means there is an obstacle in front of me and I can see it is probably a small table. Following the strategy, I can get away by first moving to the left and move forward.
Executable plans: [[3, Move leftward by 0.25], [3, Move leftward by 0.25], [0, Move forward by 0.25], [0, Move forward by 0.25]]

Environment Feedback: Step 8 is executed successfully; Step 9 is executed successfully; Step 10 is executed successfully; Step 11 is executed successfully;

Reasoning and reflection: I can see that the floorlamp is at the right side of me, slight in the front. I think it in effect of consecutively moving leftward. Also, i have got away with the obstacle. Now I need to move right and I will probably be close to the floorlamp.
Executable plans: [[2, Move rightward by 0.25], [2, Move rightward by 0.25]]]

Environment Feedback: Step 12 is executed successfully; Step 13 is executed successfully.

Reasoning and reflection: Now I can see I am almost standing right next to the floorlamp, I can complete the task.
Executable plans: [[8, Complete the current task]]

Task Success

In-context examples for EB-Manipulation

## Example 1:
Human instruction: Pick up the star and place it into the yellow container.
Input: {'object 1': [45, 13, 18], 'object 2': [72, 20, 18], 'object 3': [50, 42, 17], 'object 4': [36, 42, 18], 'object 5': [70, 43, 17], 'object 6': [50, 77, 17]}
Output:

```
{
  "visual_state_description": "From left to right, I can see a purple cylinder at
      [45, 13, 18], a blue moon at [72, 20, 18], a green star at [50, 42, 17], a
      purple cube at [36, 42, 18], a yellow container at [70, 43, 17] and a purple
      container at [50, 77, 17].",
  "reasoning_and_reflection": "The task is to pick up the star located at [50, 42,
      17] and place it into the yellow container located at [70, 43, 17]. The plan
      involves moving the gripper to the star's position, closing the gripper to
      pick it up, moving to the yellow container's position, and opening the gripper
       to release the cube into the container.",
  "executable_plan": [
    [50, 42, 17, 6, 61, 36, 1],
    [50, 42, 17, 6, 61, 36, 0],
    [47, 41, 26, 6, 61, 36, 0],
    [69, 39, 24, 88, 32, 77, 0],
    [69, 39, 24, 87, 32, 77, 1]
  ]
}
```

## Example 2:
Human instruction: Stack the red star and the olive star in sequence.
Input: {'object 1': [42, 24, 19], 'object 2': [54, 53, 19], 'object 3': [65, 68, 19], 'object 4': [39, 73, 18]}
Output:

```
{
  "visual_state_description": "From left to right, I can see a purple star at [42,
      24, 19], an olive star at [54, 53, 19], a grey star at [65, 68, 19] and a red
      star at [39, 73, 18].",
  "reasoning_and_reflection": "I need to understand the instruction first. To stack
      the objects in sequence, the red star at [39, 73, 18] should be at the bottom
      and the olive star at [54, 53, 19] should be placed on top. The plan involves
      moving the gripper to the olive star's position, closing the gripper to pick
      it up, moving to the top of the red star, and opening the gripper to release
      the olive star onto the red star.",
  "executable_plan": [
    [54, 51, 28, 3, 66, 37, 1],
    [54, 54, 20, 3, 66, 37, 0],
    [54, 51, 30, 3, 66, 37, 0],
    [39, 74, 29, 3, 66, 7, 0],
    [39, 74, 26, 3, 66, 7, 1]
  ]
}
```

In-context examples for EB-Manipulation

## Example 3:
Human instruction: Put the red star into the shape sorter.
Input: {'object 1': [65, 11, 17], 'object 2': [50, 38, 17], 'object 3': [40, 64, 37], 'object 4': [45, 83, 17]}
Output:

```
{
  "visual_state_description": "From left to right, I can see a purple star at [65,
      11, 17], a red star at [50, 38, 17], a shape sorter at [40, 64, 37] and a
      green star at [45, 83, 17].",
  "reasoning_and_reflection": "I need to identify the location of the star hole on
      the shape sorter first. I can see that the star hole is at the center of the
      shape sorter, which is at [40, 64, 37]. Then, the task is to pick up the red
      star at [50, 38, 17] and put it into the star hole on the shape sorter. The
      plan involves moving the gripper to the star's position, closing the gripper
      to pick it up, moving upwards, moving to the top of the star hole, and
      releasing the star into the hole.",
  "executable_plan": [
    [51, 38, 25, 119, 59, 26, 1],
    [51, 38, 17, 119, 59, 26, 0],
    [51, 38, 27, 119, 59, 26, 0],
    [40, 65, 49, 0, 60, 45, 0],
    [40, 65, 41, 0, 60, 45, 1]
  ]
}
```

## Example 4:
Human instruction: Wipe the horizontal area.
Input: {'object 1': [33, 36, 19], 'object 2': [52, 38, 15], 'object 3': [59, 68, 15]}
Output:

```
{
  "visual_state_description": "From left to right, I can see a yellow sponge at [33,
       36, 19], a green rectangle area at [52, 38, 15] and an orange rectangle area
      at [59, 68, 15].",
  "reasoning_and_reflection": "I need to identify which area is horizontal first. I
      can see that the orange rectangle area at [59, 68, 15] is horizontal since it
      is parallel to the green arrow representing the y-axis of the coordinate
      system annotated in the image. Then, the task is to pick up the sponge at [33,
       36, 19] and use it to wipe the horizontal area at [59, 68, 15]. The plan
      involves moving the gripper to the sponge's position, closing the gripper to
      pick it up, moving to the side of the horizontal area, and moving the sponge
      along the main direction of the horizontal area to wipe it.",
  "executable_plan": [
    [32, 34, 25, 0, 60, 34, 1],
    [32, 34, 17, 0, 60, 34, 0],
    [32, 34, 27, 0, 60, 34, 0],
    [60, 80, 18, 0, 61, 31, 0],
    [61, 54, 17, 0, 61, 31, 0]
  ]
}
```

## I.4. Output JSON Schema

Below are the JSON schemas that guide the output structure of MLLMs.

---

**Output JSON Schema for EB-ALFRED, EB-Habitat, and EB-Navigation**

```json
{
  "type": "object",
  "properties": {
    "visual_state_description": {
      "type": "string",
      "description": "Description of current state from the visual image"
    },
    "reasoning_and_reflection": {
      "type": "string",
      "description": "Summarize the history of interactions and any available
          environmental feedback. Additionally, provide reasoning as to why the last
           action or plan failed and did not finish the task."
    },
    "language_plan": {
      "type": "string",
      "description": "The list of actions to achieve the user instruction. Each
          action is started by the step number and the action name."
    },
    "executable_plan": {
      "type": "array",
      "description": "A list of actions needed to achieve the user instruction, with
           each action having an action ID and a name. Do not output an empty list."
      ,
      "items": {
        "type": "object",
        "properties": {
          "action_id": {
            "type": "integer",
            "description": "The action ID to select from the available actions given
                 by the prompt"
          },
          "action_name": {
            "type": "string",
            "description": "The name of the action"
          }
        },
        "required": ["action_id", "action_name"]
      }
    }
  },
  "required": [
    "visual_state_description",
    "reasoning_and_reflection",
    "language_plan",
    "executable_plan"
  ]
}
```

Output JSON Schema for EB-Manipulation

```
{
  "type": "object",
  "properties": {
    "visual_state_description": {
      "type": "string",
      "description": "Describe the color and shape of each object in the detection
          box in numerical order in the image. Then provide the 3D coordinates of
          the objects chosen from input."
    },
    "reasoning_and_reflection": {
      "type": "string",
      "description": "Reason about the overall plan that needs to be taken on the
          target objects, and reflect on the previous actions taken if available."
    },
    "language_plan": {
      "type": "string",
      "description": "A list of natural language actions to achieve the user
          instruction. Each language action is started by the step number and the
          language action name."
    },
    "executable_plan": {
      "type": "array",
      "description": "A list of discrete actions needed to achieve the user
          instruction, with each discrete action being a 7-dimensional discrete
          action.",
      "items": {
        "type": "object",
        "properties": {
          "action": {
            "type": "string",
            "description": "The 7-dimensional discrete action in the format of a
                list given by the prompt."
          }
        },
        "required": ["action"]
      }
    }
  },
  "required": [
    "visual_state_description",
    "reasoning_and_reflection",
    "language_plan",
    "executable_plan"
  ]
}
```

## J. Supplementary Case Studies of Successful Planning

In this section, we present successful planning examples for Claude-3.5-Sonnet, InternVL2.5-78B, GPT-4o, and Gemini-1.5-pro across EB-ALFRED, EB-Habitat, EB-Navigation, and EB-Manipulation. Refer to Figures 18, 19, 20, and 21 for detailed reasoning and planning.

**Human Instruction:** slice a tomato, heat it up in the microwave, and place it in the sink.

**1.** invalid action    **2.** find a Knife    **3.** pick up the Knife    **4.** find a Tomato    **5.** slice the Tomato    **6.** find a Sink

**7.** put down the object in hand    **8.** find a Tomato    **9.** pick up the Tomato    **10.** find a Microwave    **11.** open the Microwave    **12.** put down the object in hand

**13.** close the Microwave    **14.** turn on the Microwave    **15.** turn off the Microwave    **16.** open the Microwave    **17.** find a Tomato    **18.** pick up the Tomato

**19.** close the Microwave    **20.** find a Sink    **21.** put down the object in hand

```
Planning Step 1 (env step 1):

Visual state description: The image shows a kitchen environment with white cabinets
and a stove. There are two copper-colored pots visible on the stovetop. The
countertop appears to be white, and there's a microwave mounted above. The floor
appears to be a beige or brown color.

Reasoning & Reflection: Based on the visible kitchen environment, we need to locate
the necessary tools and ingredients first. We'll need a knife to slice the tomato,
then use the microwave that's visible in the scene, and finally locate the sink for
the final placement. The plan needs to follow the sequence of first preparing the
tomato, then heating it, and finally transferring it to its destination.
```

```
Planning Step 2 (env step 2 – env step 21):

Visual state description: The image shows a kitchen environment with white cabinets
and a stove. There are two copper-colored pots visible on the stovetop. The
countertop appears to be white, and there's a microwave mounted above. The floor
appears to be a beige or brown color.

Reasoning & Reflection: Based on the visible kitchen environment, we need to
execute a sequence of actions to slice, heat, and move a tomato. We'll need to
first locate necessary tools like a knife, then find a tomato to slice it. After
slicing, we'll need to heat it in the visible microwave, and finally transfer it to
a sink. From previous examples, I understand we need to put down tools like knives
before picking up food items for safety.
```

*Figure 18.* Planning example of Claude-3.5-Sonnet in EB-AFRED.

**Human Instruction:** Move the spoon to the brown table, the sponge to the brown table, and the cleanser to the black table.

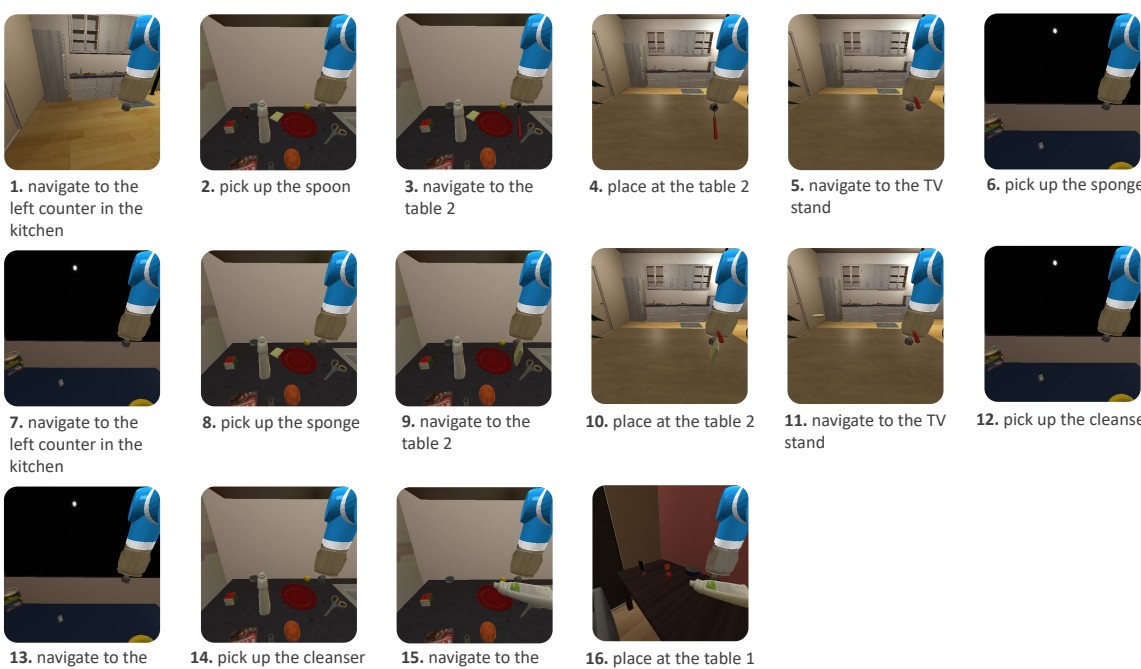

**Planning Step 1 (env step 1 – env step 6):**

**Visual state description:** The image shows a kitchen with wooden flooring, a refrigerator, a sink, and several cabinets. There are no visible objects such as a spoon, sponge, or cleanser in the immediate view.

**Reasoning & Reflection**: The task requires moving a spoon, sponge, and cleanser to specific tables. Since these objects are not visible in the current view, the robot needs to navigate to locations where these objects might be found. The robot will start by navigating to the left counter in the kitchen to search for the spoon.

**Planning Step 2 (env step 7 – env step 12):**

**Visual state description:** The image shows a room with a blue carpet, a can, and a small object on the floor. There is a cabinet and a table visible in the scene.

**Reasoning & Reflection**: The robot has successfully moved the spoon to the brown table (table 2). However, the attempt to pick up the sponge from the TV stand was unsuccessful because the sponge was not near the robot. The robot needs to locate the sponge and the cleanser to complete the task.

**Planning Step 3 (env step 13 – env step 16):**

**Visual state description:** The image shows a room with a dark blue floor and a black wall. There is a can on the floor and a small object near it. The robot arm is visible in the upper right corner of the image.

**Reasoning & Reflection**: The robot has successfully moved the spoon and the sponge to the brown table (table 2). However, the last attempt to pick up the cleanser from the TV stand was unsuccessful because the cleanser was not near the robot. The robot needs to locate the cleanser before attempting to pick it up.

*Figure 19.* Planning example in EB-Habitat for InternVL2.5-78B.

**Human Instruction:** navigate to the Pot in the room and be as close as possible to it.

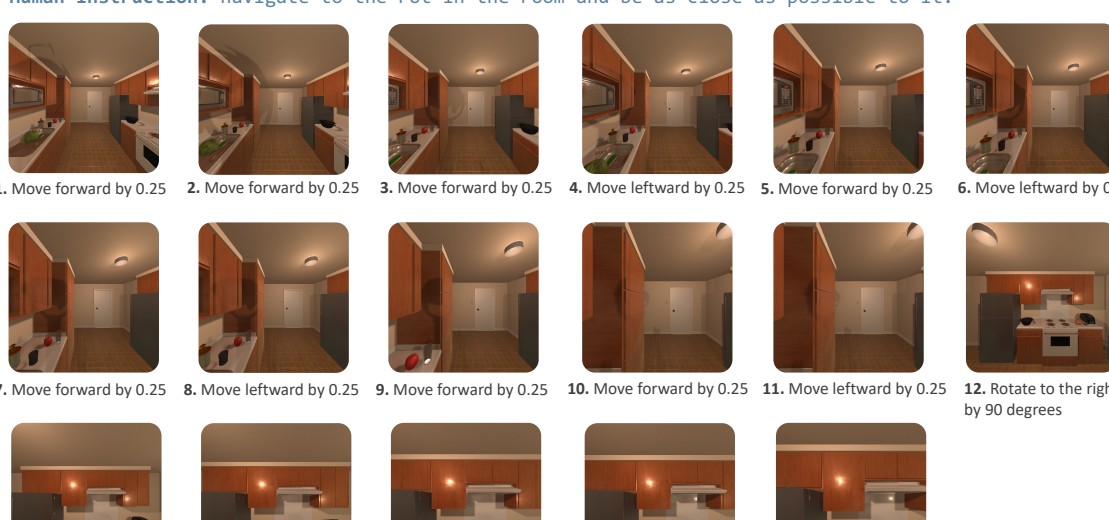

**1.** Move forward by 0.25  **2.** Move forward by 0.25  **3.** Move forward by 0.25  **4.** Move leftward by 0.25  **5.** Move forward by 0.25  **6.** Move leftward by 0.25

**7.** Move forward by 0.25  **8.** Move leftward by 0.25  **9.** Move forward by 0.25  **10.** Move forward by 0.25  **11.** Move leftward by 0.25  **12.** Rotate to the right by 90 degrees

**13.** Move forward by 0.25  **14.** Move forward by 0.25  **15.** Move forward by 0.25  **16.** Move leftward by 0.25  **17.** Move forward by 0.25

---

**Planning Step 1 (env step 1 – env step 2):**

The pot is located on the countertop to the front left of my current position. I can approach it by moving forward and leftward. I will start with moving forward towards the pot since the path ahead seems clear."

---

**Planning Step 2 (env step 3 – env step 5):**

I moved forward successfully, which brought me closer to the pot. The pot is now more directly in front and slightly to the left. The approach has been effective so far, following the planned movements, without any obstacles blocking my path.

---

**Planning Step 3 (env step 6):**

I've successfully moved closer to the pot. It's now directly in front of the left countertop but I haven't reached the closest proximity possible. Continuing to adjust my position slightly leftward and forward will get me as close as possible.

---

**Planning Step 4 (env step 7 – env step 8):**

I've moved forward and left to get closer to the pot, but my last left movement was blocked by the baseboard. This suggests I'm near an edge or obstacle. I can still move forward to get closer to the pot since it's in front.

---

**Planning Step 5 (env step 9 – env step 11):**

I managed to move forward but my sideward movements were blocked by the baseboard, indicating a limitation at this spot. Further approaches will be more successful by moving forward and then attempting lateral moves to align with the pot.

---

**Planning Step 6 (env step 12 – env step 16):**

I've moved forward several times towards where the pot was located. Despite attempts to move left to get closer, these actions were repeatedly blocked by obstacles. This indicates a misalignment with the countertop or edge preventing approach. Since I've lost sight of the pot, rotation is required.

---

**Planning Step 7 (step 17):**

After rotating, the pot appeared on the counter. I've moved forward and then left to approach it. I can now see it clearly in my current position, slightly ahead.

---

*Figure 20.* Planning example of GPT-4o in EB-Navigation.

**Human Instruction:** Stack the front star on top of the right cylinder.

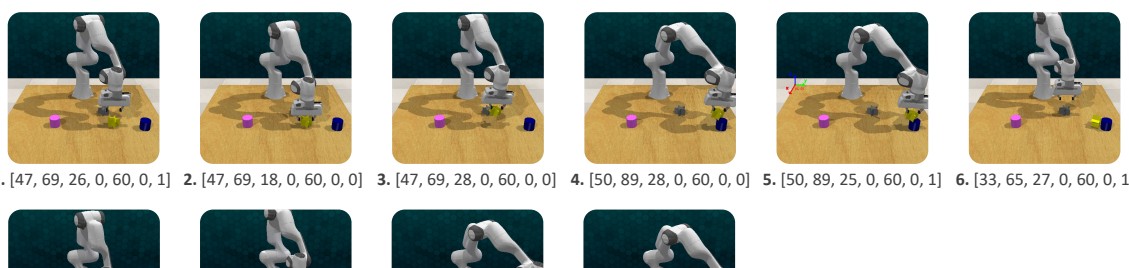

**1.** [47, 69, 26, 0, 60, 0, 1]   **2.** [47, 69, 18, 0, 60, 0, 0]   **3.** [47, 69, 28, 0, 60, 0, 0]   **4.** [50, 89, 28, 0, 60, 0, 0]   **5.** [50, 89, 25, 0, 60, 0, 1]   **6.** [33, 65, 27, 0, 60, 0, 1]

**7.** [33, 65, 19, 0, 60, 0, 0]   **8.** [33, 65, 29, 0, 60, 0, 0]   **9.** [51, 90, 29, 0, 60, 0, 0]   **10.** [51, 90, 21, 0, 60, 0, 1]

---

**Planning Step 1 (env step 1 – env step 5):**

**Visual state description:** From left to right, I can see a magenta cylinder at [47, 31, 20], a gray gear at [33, 64, 19], a yellow star at [47, 69, 18], and a navy cylinder at [50, 89, 20].

**Reasoning & Reflection:** The instruction is to stack the front star on top of the right cylinder. The yellow star at [47, 69, 18] should be picked up and placed on the navy cylinder at [50, 89, 20]. The plan involves moving the gripper to the yellow star's position, closing the gripper to pick it up, moving to the top of the navy cylinder, and opening the gripper to release the yellow star onto the navy cylinder.

---

**Planning Step 2 (env step 6 – env step 10):**

**Visual state description:** From left to right, I can see a magenta cylinder at [47, 31, 20], a gray star at [33, 65, 19], a yellow star at [50, 88, 25], and a blue cylinder at [51, 90, 20].

**Reasoning & Reflection:** The task is to stack the front star on top of the right cylinder. The front star is the gray star at [33, 65, 19], and the right cylinder is the blue cylinder at [51, 90, 20]. The plan involves moving the gripper to the gray star's position, closing the gripper to pick it up, moving to the top of the blue cylinder, and opening the gripper to release the gray star onto the blue cylinder.

---

*Figure 21.* Planning example of Gemini-1.5-pro in EB-Manipulation.

# K. Supplementary Case Studies of Unsuccessful Planning

We also present failure cases across different tasks, highlighting key challenges in perception, reasoning, and planning. In EB-ALFRED, we showcase a planning error where the model fails to generate an effective executable plan to accomplish the task. In EB-Manipulation, we analyze a perception error where the agent misidentifies key objects, leading to incorrect execution. In EB-Navigation, we illustrate a reasoning error where the model struggles to interpret spatial relationships. These examples provide insights into the limitations of current models and highlight areas for improvement in object recognition, planning, and spatial reasoning. Refer to Figures 22, 23, and 24 for details.

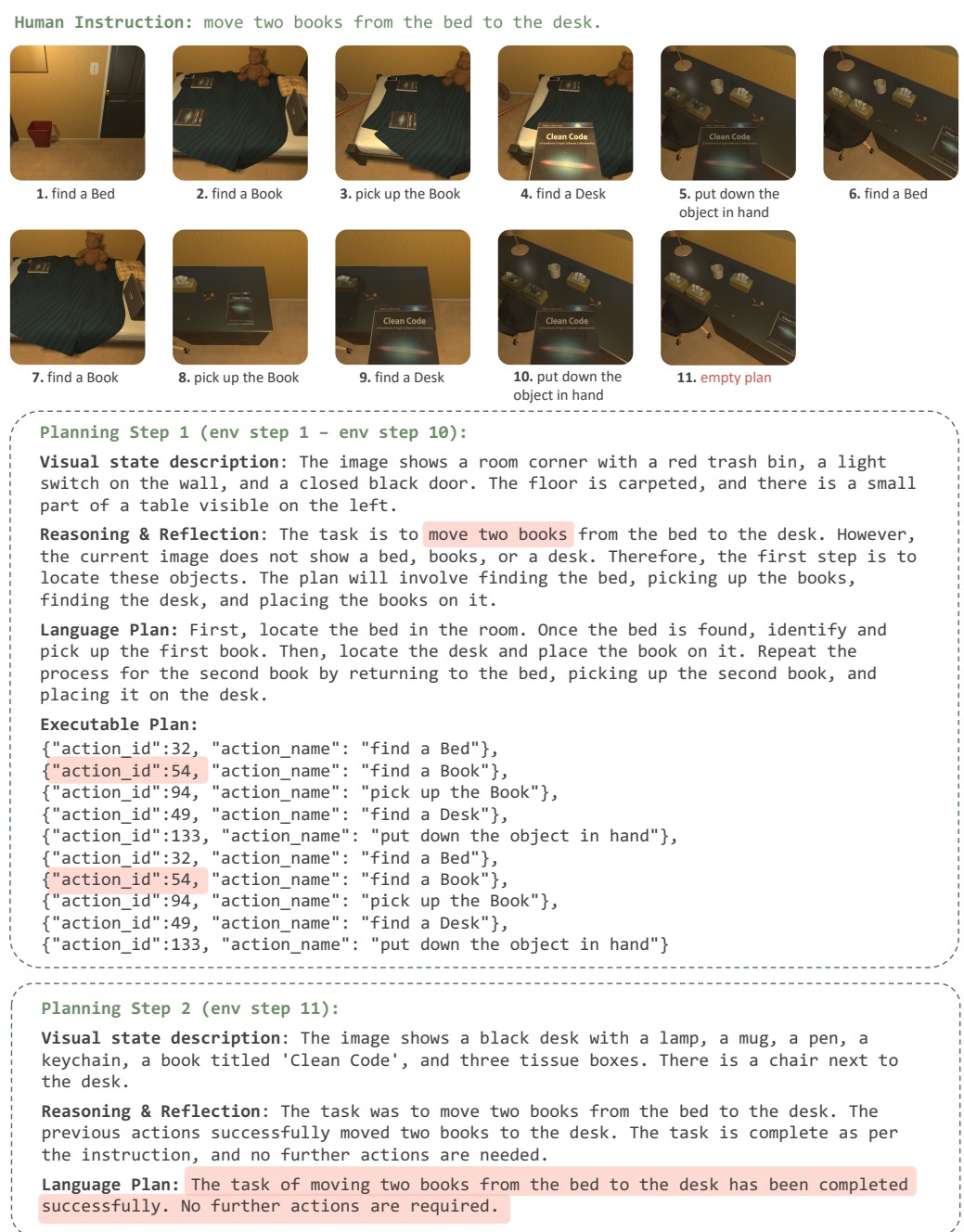

*Figure 22.* Planning Error Example in EB-ALFRED: The agent was supposed to locate "Book_2" by the 7th action but instead continued interacting with the first book.

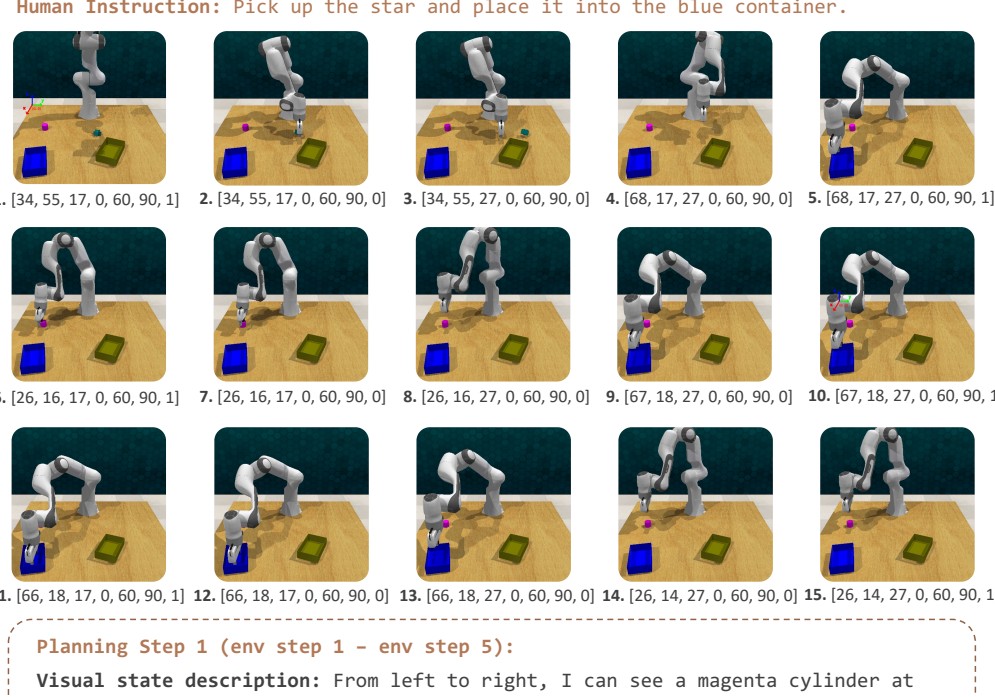

**Human Instruction:** Pick up the star and place it into the blue container.

1. [34, 55, 17, 0, 60, 90, 1]  2. [34, 55, 17, 0, 60, 90, 0]  3. [34, 55, 27, 0, 60, 90, 0]  4. [68, 17, 27, 0, 60, 90, 0]  5. [68, 17, 27, 0, 60, 90, 1]

6. [26, 16, 17, 0, 60, 90, 1]  7. [26, 16, 17, 0, 60, 90, 0]  8. [26, 16, 27, 0, 60, 90, 0]  9. [67, 18, 27, 0, 60, 90, 0]  10. [67, 18, 27, 0, 60, 90, 1]

11. [66, 18, 17, 0, 60, 90, 1]  12. [66, 18, 17, 0, 60, 90, 0]  13. [66, 18, 27, 0, 60, 90, 0]  14. [26, 14, 27, 0, 60, 90, 0]  15. [26, 14, 27, 0, 60, 90, 1]

---

**Planning Step 1 (env step 1 – env step 5):**

**Visual state description:** From left to right, I can see a magenta cylinder at [26, 16, 18], a blue container at [68, 17, 17], a green star at [34, 55, 17], and an olive container at [57, 63, 17].

**Reasoning & Reflection:** The task is to pick up the star located at [34, 55, 17] and place it into the blue container located at [68, 17, 17]. The plan involves moving the gripper to the star's position, closing the gripper to pick it up, moving to the blue container's position, and opening the gripper to release the star into the container.

---

**Planning Step 2 (env step 6 – env step 10):**

**Visual state description:** From left to right, I can see a pink star at [26, 16, 17], a blue container at [67, 18, 17], and a green container at [57, 63, 17].

**Reasoning & Reflection:** The task is to pick up the star located at [26, 16, 17] and place it into the blue container located at [67, 18, 17]. The plan involves moving the gripper to the star's position, closing the gripper to pick it up, moving to the blue container's position, and opening the gripper to release the star into the container.

---

**Planning Step 3 (env step 11 – env step 15):**

**Visual state description:** From left to right, I can see a blue container at [26, 14, 17], a purple star at [66, 18, 17], and a green container at [57, 63, 17].

**Reasoning & Reflection:** The task is to pick up the star located at [66, 18, 17] and place it into the blue container located at [26, 14, 17]. The plan involves moving the gripper to the star's position, closing the gripper to pick it up, moving to the blue container's position, and opening the gripper to release the star into the container.

---

*Figure 23.* Perception Error Example in EB-Manipulation: the agent erroneously observed the color of the object.

**Human Instruction:** navigate to the Pot in the room and be as close as possible to it.

**1.** Move forward for 0.25

**2.** Move forward for 0.25

**3.** Move forward for 0.25

**4.** Move forward for 0.25

**5.** Move forward for 0.25
Invalid

**6.** Move leftward for 0.25
Invalid

**7.** Move forward for 0.25
Invalid

**8.** Move leftward for 0.25
Invalid

**9.** Move leftward for 0.25
Invalid

**10.** Tilt the camera upward by 30 degrees

**11.** Move leftward for 0.25
Invalid

**12.** Tilt the camera downward by 30 degrees

**13.** Move backward for 0.25

**14.** Move rightward for 0.25

**15.** Rotate to the right by 90 degrees

**16.** Move forward for 0.25

**17.** Rotate to the left by 90 degrees

**18.** Rotate to the left by 90 degrees

**19.** Move forward for 0.25

**20.** Move forward for 0.25

---

**Planning Step 1 (env step 1 – env step 2):**

My goal is to locate the pot in this kitchen. Given the current view, there's no pot visible directly. I should move forward to explore more of my surroundings, as moving forward will help me view more of the countertop and other areas not fully visible right now.

⋮

**Planning Step 4 (env step 6):**

The previous action sequence of moving forward eventually led to an invalid move due to being blocked by the countertop. This indicates proximity to the target. The pot is now in front and a bit to the left, requiring leftward and forward movements to get closer.

⋮

**Planning Step 13 (env step 19 – env step 20):**

The pot is visible now, slightly to the right. Previous attempts resulted in being blocked by the counter. In order to move closer, we should navigate around obstacles and ensure our path aligns more effectively towards the target, adjusting the direction as needed to avoid obstructions.

---

*Figure 24.* Reasoning Error Example in EB-Navigation: the agent recognized it was blocked by the countertop but failed to attempt navigating around it.

