# OpenReview forum: "EmbodiedBench: Comprehensive Benchmarking Multi-modal Large Language Models for Vision-Driven Embodied Agents"
_ICML.cc/2025/Conference — ICML 2025 oral_

### Official Review · Reviewer_MUiH · 2025-03-11

**Overall Recommendation:** 5

**Summary:**

The paper proposes a powerful and comprehensive benchmark, EmbodiedBench, for both high-level and low-level actions in embodied intelligence. It consists of four distinct subdatasets, ranging from high-level semantic tasks to low-level metric tasks, with each subdataset having its own focus. To build the entire benchmark, the authors first collect results from other datasets and utilize a simulator for data generation. They then correct and refine the limitations of the original datasets. Moreover, the tasks are categorized into six types, covering common and challenging embodied intelligence tasks. Additionally, evaluations conducted on both proprietary and open-source models demonstrate that this dataset presents significant challenges.

**Claims And Evidence:**

None.

**Essential References Not Discussed:**

None.

**Experimental Designs Or Analyses:**

1. In Task Planner, how accurate is it for the agent to execute multiple steps at once? Is there an ablation study comparing this with single-step execution?
2. For multi-image inputs, I understand that due to the limitations of large multimodal models, directly adding multiple images might lead to suboptimal results. Is there a comparative experiment that converts the information from multiple images into text and provides it to the model in an in-context manner?

**Methods And Evaluation Criteria:**

1. In EB-Manipulation, I am a bit confused about the necessity of providing the detection box to the MLLM. (1) In what manner is the box provided to the MLLM? Is it drawn directly on the image, and have other forms been considered? (2) Does the color of the box or the thickness of the box's outline impact the enhancement effect? (3) Why does adding the box lead to a decrease in performance for the navigation subtask? I think it would be helpful to analyze this further in relation to the task's inputs and outputs. I saw some discussion about the detection box in the supplementary materials, but it did not fully resolve my confusion.
2. Please explain the differences between high-level and low-level trajectories. Are there differences in difficulty or the way instructions are expressed?
3. What is the effectiveness of using ChatGPT directly to generate data? How can accuracy be verified? Is there a human review process involved? If I believe that ChatGPT may have difficulty understanding spatial concepts (as referenced in the final evaluation table), how can we ensure the accuracy of the data it generates?

**Other Comments Or Suggestions:**

I suggest including the evaluation of the qwen2.5-VL 7B and 72B models in the final version of the paper.

**Other Strengths And Weaknesses:**

Weakness1: The entire benchmark is centered around scenarios in a simulator, but does not address potential limitations that may be encountered in real-world scenarios.

**Questions For Authors:**

None.

**Relation To Broader Scientific Literature:**

None.

**Theoretical Claims:**

None.

---

> ### Author Rebuttal · Authors · 2025-04-01
>
> Thank you for reviewing our work and providing valuable feedback. We have carefully addressed your concerns below. Please let us know if you have any further questions.
>
> The anounymous link for figures is https://anonymous.4open.science/r/rebuttal-3568/rebuttal_file.pdf. We use "4o" to refer to GPT-4o and "Claude" to refer to Claude-3.5-Sonnet.
>
> **Q1: Detection box in EB-Manipulation. (1) How are they provided to MLLMs? (2) Do color or thickness impact performance? (3) Why does performance drop in EB-Navigation?**
>
> **A1:** (1) Detection boxes are directly drawn on images to guide the model’s focus on relevant regions. An example is on the right side of **Figure 1 in the anonymous link**.
>
> (2) Our additional ablation studies on EB-Manipulation base subset show:
> - All detection boxes improve performance over no box.
> - Box color has minimal impact; even the color similar to desk (yellow) only causes a slight drop (4.2%).
> - Increasing thickness (1px->2 or 3px) slightly reduces performance (<6.2%).
>
> |  | Default (red, 1px) | 2px  | 3px  |  black  | yellow  | no Box |
> | - | - | - | - |- | - | - |
> |  Claude | 37.5 | 31.3 | 33.3 | 37.5 | 33.3 | 29.2  |
>
> (3) **Figure 2 in the anonymous link** shows that in EB-Navigation, multiple detection boxes can obscure distant objects, reducing visibility. In contrast, EB-Manipulation keeps objects at a fixed distance, ensuring clear visibility. As a result, detection boxes affect the two environments differently.
>
> In EB-Navigation, we tested using **a single box only on the target object** that can reduce obstruction, showing consistently improved accuracy. To better reflect real-world scenarios, EB-Navigation omits detection boxes by default, requiring the MLLM agent to detect and recognize objects.
>
> | Model | No Box | One Box | Multi Box |
> |-|-|-|  -|
> | 4o | 61.7 | 68.3 | 53.3  |
> | Claude | 46.7 | 58.3 | 48.3  |
>
> **Q2: Differences between high-level and low-level trajectories. Do they vary in difficulty or instructions?**
>
> **A2:** "High-level" and "low-level" refer to different action representations based on their executability in robotic systems (see Section 3, Paragraph 1). **The trajectory structure and instructions are consistent across all high-level and low-level tasks; the key difference lies in how actions are represented—either as high-level abstractions or low-level primitives.**
>
> In terms of difficulty, our results show that low-level tasks are more challenging for MLLM agents. This is because they require stronger perception and spatial awareness, which remain limitations for current MLLMs.
>
> **Q3: How effective is ChatGPT in data generation? How is accuracy verified? Is there a human review process? How do we ensure the accuracy of the generated data?**
>
> **A3**: **Our dataset preparation combines GPT-4o and human annotation to ensure high quality**. GPT-4o is used not for full data generation but to enhance linguage instruction diversity.
>
> For example, in EB-ALFRED, task descriptions (PDDL) and instructions for the base subset are sampled from ALFRED. For other subsets (e.g., "Common Sense"), we craft 10 examples to guide GPT-4o in generating augmented instructions. To ensure accuracy, we manually review all instructions for correctness and coherence with PDDL descriptions, revising or discarding invalid data. This human-in-the-loop approach ensures dataset reliability.
>
> **Q4: How accurate is the multi-step planner? An ablation study comparing it with single-step execution.**
>
> **A4:** The multi-step planner is crucial for improving performance while reducing API/inference costs. To assess its impact, we compared multi-step and single-step execution. Results show significant performance drops with single-step execution on EB-ALFRED base subset. These results confirm the importance of multi-step planning.
>
> |  | Default | Single Step |
> |-|-|-|
> | 4o   | 64 | 36 |
> | Claude    | 72 | 62 |
>
> **Q5: Evaluation of converting multi-step images into text as in-context information.**
>
> **A5:** We tested incorporating multi-step observation descriptions into the context. While this method did not improve GPT-4o’s performance, it led to a 4% gain for Claude-3.5 on EB-ALFRED (Base). We plan to offer this as an optional feature in our code release.
>
> |  | Default | w/ Image Descriptions |
> |-|-|-|
> | 4o | 64 | 64 |
> | Claude | 72 | 76 |
>
> **Q6: Limitations of simulation in real-world scenarios.**
>
> **A6:** We acknowledge the limitations of simulations in capturing real-world challenges. Please refer to **Q1 of Reviewer yZa7** for a detailed discussion. We will add a discussion on the limitations in the revision.
>
> **Q7: Evaluation of qwen2.5-VL models.**
>
> **A7:** We evaluated Qwen2.5-VL models on EmbodiedBench and observed notable improvements over Qwen2-VL. Qwen2.5-VL-72B achieves an overall score of 34.7, surpassing previous open-source SOTA, InternVL2.5-78B (33.9). We will include evaluations of more recently released MLLMs in our updated manuscript.

---

> > ### Comment · Reviewer_MUiH · 2025-04-03
> >
> > Thanks for the rebuttal, the replies address my primary concerns effectively. So I raise my rating as strong accept.

---

### Official Review · Reviewer_R7Zw · 2025-03-13

**Overall Recommendation:** 4

**Summary:**

This paper introduces EmbodiedBench, a comprehensive benchmark for evaluating vision-driven embodied agents based on multi-modal large language models (MLLMs). The benchmark features 1,128 testing instances across four environments, covering both high-level semantic tasks and low-level atomic actions, with six meticulously curated subsets evaluating essential agent capabilities such as common sense reasoning, complex instruction following, spatial awareness, visual perception, and long-term planning. Through extensive experiments, the authors evaluate 13 leading proprietary and open-source MLLMs within EmbodiedBench, revealing that MLLMs excel at high-level tasks but struggle with low-level manipulation, with the best model, GPT-4o, scoring only 28.9% on average.

**Claims And Evidence:**

Yes

**Essential References Not Discussed:**

No.

**Experimental Designs Or Analyses:**

Yes

**Methods And Evaluation Criteria:**

Yes

**Other Comments Or Suggestions:**

See Weaknesses

**Other Strengths And Weaknesses:**

Strengths:

1. EmbodiedBench is the first benchmark to comprehensively evaluate MLLM-based embodied agents across multiple environments and task levels, providing a standardized platform for comparison.

2. The benchmark includes six capability-oriented subsets that allow for detailed analysis of different agent capabilities, offering valuable insights into model limitations.

3. The authors develop a unified agent framework that effectively integrates egocentric visual perception, few-shot in-context examples, interaction history, and environment feedback for decision-making.

4.  The paper presents thorough experiments with 13 state-of-the-art MLLMs, providing valuable insights into their performance on various embodied tasks.

Weaknesses:

1.  The absence of evaluation in real-world physical environments. The entire benchmark is implemented in a virtual environment, which raises the question of whether the results of the virtual benchmark measurements reflect the capabilities of the model in the real world. Embodied intelligence is largely meant to operate in the real world, and I would suggest that the authors add some real-world experiments, or some discussion.

2. The review includes a very large number of MLLMs, but the VLA (Vision-Language-Action) model is missing. I suggest adding some experiments.

**Questions For Authors:**

No

**Relation To Broader Scientific Literature:**

This paper significantly advances the evaluation of multimodal macromodels

**Theoretical Claims:**

N/A

---

> ### Author Rebuttal · Authors · 2025-04-01
>
> Thank you for reviewing our work and providing valuable feedback. We have carefully addressed your concerns below. Please let us know if you have any further questions.
>
> **Q1: The absence of evaluation in real-world physical environments ... Embodied intelligence is largely meant to operate in the real world, and I would suggest that the authors add some real-world experiments, or some discussion.**
>
> **A1:** We agree with the reviewer on the importance of real-world evaluation. However, there is an inherent trade-off between reproducibility, cost, safety, and real-world applicability. While real-world testing is crucial for practical deployment, simulated benchmarks provide a standardized and easily reproducible environment, reducing the time, financial burden, and safety risks associated with real-world evaluation [1,2]. EmbodiedBench is a step forward in enabling evaluating MLLM agents in diverse simulated embodied tasks. Future research could benefit from more realistic and complex embodied simulations [3] or standardized and cost-effective real-world test suites [4,5]. **In our revision, we will add a discussion on this limitation at the end of the main paper.**
>
> [1] Evaluating Real-World Robot Manipulation Policies in Simulation. CoRL 2024.
>
> [2] Visualagentbench: Towards large multimodal models as visual foundation agents. ICLR 2025.
>
> [3] Behavior-1k: A human-centered, embodied ai benchmark with 1,000 everyday activities and realistic simulation. ArXiv, 2024
>
> [4] Learning Fine-Grained Bimanual Manipulation with Low-Cost Hardware. RSS 2023.
>
> [5] Mobile aloha: Learning bimanual mobile manipulation using low-cost whole-body teleoperation. CORL 2024.
>
> **Q2: The review includes a very large number of MLLMs, but the VLA model is missing. I suggest adding some experiments.**
>
> **A2:** We appreciate the reviewer’s suggestion. **We reviewed VLAs in paragraph 3, Appendix A**. While VLA models such as Octo, OpenVLA, and RDT-1B have shown strong performance in manipulation tasks, there is currently no open-source VLA model that can handle both high-level household tasks and low-level navigation & manipulation simultaneously. Consequently, there is no directly comparable VLA model for our evaluation.
>
> However, to explore their capabilities, we conducted experiments on three pretrained VLA models (Octo, OpenVLA, and RDT-1B) using the EB-Manipulation (Base) subset. All models achieved a **0% success rate**. This outcome can be attributed to the distribution shift: existing robotic foundation models [6][7] are primarily trained on real-world datasets such as Open X-Embodiment. The large domain gap between these datasets and our simulator environment prevents these models from generalizing effectively without fine-tuning. This reinforces the need for a more general framework, as proposed in our paper, that enables adaptation to diverse tasks without fine-tuning.
>
> While a few VLA models [8][9] have been developed for the same simulator we use, they are not publicly available. Furthermore, other available models [10][11] have input formats that differ significantly from our environment (e.g., proprioception mismatch), making direct evaluation infeasible. The only available VLA model trained in the same simulator as ours is 6D-CLIPort [12], which processes multi-view observations and language instructions to generate 6-DoF actions. Below are the evaluation results on EB-Manipulation:
>
> | EB-Manipulation | Base  | Common |   Complex  | Spatial   | Visual | Avg |
> | - | - | - |- | - | - |- |
> |  6D-CLIPort   |   8.3  |  6.3  | 2.1 |  16.7  | 16.7 | 10.0 |
>
> The model performs well on tasks that rely on visual understanding and spatial reasoning. However, its success rate drops significantly on tasks requiring common sense reasoning or understanding complex instructions, suggesting that action-grounded VLA models may struggle in these areas. In contrast, our MLLM-based agent framework achieves higher performance, with GPT-4o reaching an average score of 28.9. This result highlights the effectiveness of our MLLM agent framework and the potential of general MLLMs as embodied agents.
>
> [6] OpenVLA: An Open-Source Vision-Language-Action Model. CORL 2024.
>
> [7] RDT-1B: a Diffusion Foundation Model for Bimanual Manipulation, ICLR 2025.
>
> [8] MoLe-VLA: Dynamic Layer-skipping Vision Language Action Model via Mixture-of-Layers for Efficient Robot Manipulation. Arxiv 2025.
>
> [9] HAMSTER: Hierarchical Action Models For Open-World Robot Manipulation, ICLR 2025.
>
> [10] RVT-2: Learning Precise Manipulation from Few Examples. RSS 2024.
>
> [11] SAM2Act: Integrating Visual Foundation Model with A Memory Architecture for Robotic Manipulation. Arxiv 2025.
>
> [12] VLMbench: A Compositional Benchmark for Vision-and-Language Manipulation, NeurIPS 2022.

---

### Official Review · Reviewer_k7jC · 2025-03-14

**Overall Recommendation:** 4

**Summary:**

This paper proposes embodiedbench, a benchmark for evaluating MLLMs' capability in a diverse set of embodied tasks. Specifically, the tasks range from high-level semantic tasks to low-level tasks with atomic actions. Furthermore, the tasks under different simulators are classified into different subsets, to evaluate agents' capability in common sense reasoning, complex instruction following, spatial awareness, visual perception and long-term planning. This paper benchmarks the performance of multiple open-source MLLMs and close-source MLLMs for embodied tasks, providing interesting insights in how MLLMs perform at high-level tasks and low-level manipulation.

**Claims And Evidence:**

Strengths:
1. This paper proposes a benchmark for evaluating MLLM's capability in diverse set of embodied tasks, covering a wider range of tasks compared with previous research.
2. The design of including both high-level semantics and low-level tasks are essential, which is also supported with the large performance difference observed for MLLMs.
3. Further classifying the agents' capability into different categories can bring more insights while doing evaluation.

**Essential References Not Discussed:**

N/A

**Experimental Designs Or Analyses:**

Strengths:
1. Detailed ablations for both language-centric analysis and vision-centric analysis.
2. Interesting findings indicating that vision is crucial for embodied tasks with low-level actions.

**Methods And Evaluation Criteria:**

Strengths:
1. Benchmarking both open-source MLLMs and close-source MLLMs on the proposed embodiedbench, which serves as a good start point for futher work.

Weakness:
1. MLLMs performance might relate to the prompt being used, and one single prompt is used for all the models. Exploring how MLLMs work with a different set of prompts might bring more insights and robust conclusion about how well different MLLM work for embodied tasks. Some ablations for the language prompt provided in Sec.5.3 can partially mitigate this issue.

**Other Comments Or Suggestions:**

N/A

**Other Strengths And Weaknesses:**

N/A

**Questions For Authors:**

N/A

**Relation To Broader Scientific Literature:**

The proposed benchmark can be very crucial for future development of MLLMs for embodied AI tasks, especially in low-level manipulation category.

**Theoretical Claims:**

N/A

---

> ### Author Rebuttal · Authors · 2025-04-01
>
> Thank you for reviewing our work and providing valuable feedback. We have carefully addressed your concerns below. Please let us know if you have any further questions.
>
>
> **Q1: MLLMs performance might relate to the prompt being used, and one single prompt is used for all the models. Exploring how MLLMs work with a different set of prompts might bring more insights and robust conclusion about how well different MLLM work for embodied tasks. Some ablations for the language prompt provided in Sec.5.3 can partially mitigate this issue.**
>
>
> **A1:** We agree that prompt design plays an important role in MLLM agent performance. In Section 5.3, we analyzed the effects of textual environmental feedback and the number of in-context examples. To further investigate prompt robustness, we conducted additional studies on the EB-ALFRED "Base" subset, evaluating the following variations:
>
> 1. **"No Guideline"** – Removing instructional guidelines intended to assist model generation (see Page 19).
> 2. **"Prompt v2"** – Rewriting the original prompts using GPT-4o to rephrase and restructure while preserving similar information.
>
> The results show that removing guidelines has no effect on model performance, while "Prompt v2" causes a slight 4% performance drop. **This suggests that our MLLM agent is relatively robust to prompt modifications** compared to its sensitivity to environmental feedback (around 10% drop) and in-context examples (more than 20% drop).
>
>
> |  | GPT-4o  |GPT-4o (no guideline)   |  GPT-4o (prompt v2)  | Claude-3.5-Sonnet  | Claude-3.5-Sonnet (no guideline)  |  Claude-3.5-Sonnet (prompt v2) |
> | -------- | -------- | -------- |-------- | -------- |-------- | -------- |
> |  EB-ALFRED (Base)   |   64  |  64  | 60 | 72 |   72  |68 |
>
>
> Furthermore, we also examined the impact of reasoning in in-context examples (i.e., ReAct prompting [1]). Removing this reasoning step leads to an 8% performance drop for GPT-4o and a 4% drop for Claude-3.5-Sonnet, indicating that reasoning within in-context examples is more impactful than prompt rephrasing or guidelines, though still secondary to environmental feedback and in-context examples.
>
> |  | GPT-4o  |GPT-4o (w/o ReAct)   |   Claude-3.5-Sonnet  | Claude-3.5-Sonnet (w/o ReAct)   |
> | -------- | -------- | -------- |-------- | -------- |
> |  EB-ALFRED (Base)   |   64  |  56  | 72 |  68  |
>
> In summary, our MLLM agent framework is robust to prompt rephrasing and guideline removal. However, environmental feedback and in-context examples, as discussed in Section 5.3, have a much greater impact on performance.
>
> [1] React: Synergizing reasoning and acting in language models. ICLR, 2023.

---

### Official Review · Reviewer_yZa7 · 2025-03-14

**Overall Recommendation:** 5

**Summary:**

The authors present EmbodiedBench - a set of diverse tasks and environment to evaluate MLLMs for embodied agents. They present high-level benchmark environments - EB-Habitat, EB-ALFRED and  low-level benchmark environments - EB-Navigation and EB-Manipulation. The tasks are also divided into basic task solving, common sense, complex instructions, spatial awareness, visual perception and long-term planning.

They evaluate 13 different MLLMs on this benchmark, showing that high-level tasks are easier than lower-level tasks, and that visual cues are more important for low-level tasks than higher-level tasks. They perform ablations showing image resolution should be reasonable, multi-step images harm the MLLMs, and visual in-context learning helps. They also perform error analysis showing the reasons for failures in specific aspects of robotics (planning, perception, reasoning).

**Claims And Evidence:**

1. **EmbodiedBench is a useful tool for evaluating MLLMs in embodied settings**: The results are very useful and they help use understand the performance of difference LLMs in different settings and types of tasks. The insights are interesting and would help further research on this domain. One specific issue that I find with this approach is that all data is in simulation, and there are no real-world episodes, or evaluations. It is hard to say how aligned with the real-world performance are the results.
2. **MLLMs excel at high-level, struggle at low-level tasks**: This claim is shown by the performance on EB-ALFRED and EB-Habitat vs EB-Navigation and EB-Manipulation. I believe this is an interesting insight.
3. **Visual cues are necessary for low-level tasks:** This makes sense, since precision is key in such tasks. It is interesting that the authors empirically find this for MLLMs using their benchmark.

**Essential References Not Discussed:**

Can not think of anything that might be missing. They consider a good amount of related benchmarks and show a comparison table.

**Experimental Designs Or Analyses:**

- Agent Design: The agent design uses single-step images for efficiency and provide valid skill sets for each task to the MLLM. For manipulation, they provide markers and boxes. I think this is a fair design, considering MLLMs struggle with multi-step input.
- Task Planning: This design is particularly interesting to me. Instead of doing per-step planning, they plan multiple steps in a single go and let the MLLM decide the number of steps. How do the authors prevent failure of plans? Are there any qualitative examples of this? It would be nice to discuss this if possible.
- Ablations: I think the ablations are also careful and evaluate several aspects of the approach. The results show importance of feedback, in-context learning, camera resolution, detection boxes (for manipulation), multi-step input, visual in-context learning.
- Error analysis: They show interesting error analysis and show subtypes of error, along with discussion on why different LLMs might fail on different subtypes.

**Methods And Evaluation Criteria:**

1. They create a benchmark over different environments and tasks to evaluate MLLMs for embodied AI. I think this is a sound approach and casts a wide-net for MLLM evaluation.
2. They describe the data creation in detail, and it is well-grounded in prior works. The task subsets are well thought.

**Other Comments Or Suggestions:**

- Line 215: Incorrect quotes around "navigate"
- Line 257, Col 2: "constained"-> contained or constrained?
- Line 306, Col 2: Incorrect quotes around "Lang"
- Line 386: Mentions "multi-view", which is in ablation. Have the authors tried using panoramic images from multi-view?
- Is it possible to train some e2e or modular policies on this benchmark and show how they perform on this benchmark? Might be interesting to compare the best MLLM with such policies. I agree that this is not a necessity since the purpose of the benchmark is to evaluate MLLMs.
- Section 6 title should be: "Conclusion"

**Other Strengths And Weaknesses:**

The paper is pretty interesting overall, has nice insights and analysis. It adds value to the community, and might be helpful in evaluating future MLLMs. It would be nice to have some real-world evaluations, or something that shows that the benchmark performance translates.

**Questions For Authors:**

1. How is the complex instruction understanding task different from long-horizon? Is there an intersection?
2. How is the data generated for EB-Navigation? Which Python program what used? I did not find this in appendix.

**Relation To Broader Scientific Literature:**

The paper presents an interesting benchmark to evaluate MLLMs for Embodied AI agents and is one of the first ones to do so. The insights are pretty interesting and the evaluation/analysis is comprehensive. I think this will help understand MLLMs better for robotics, especially since it is a up-and-coming field.

**Theoretical Claims:**

N/A

---

> ### Author Rebuttal · Authors · 2025-04-01
>
> Thank you for reviewing our work and providing valuable feedback. We have carefully addressed your concerns below. Please let us know if you have any further questions.
>
> **Q1: No real-world evaluations. It is hard to say how aligned with the real-world performance**
>
> **A1:** We agree with the reviewer regarding the gap between simulated and real-world tasks. While real-world evaluation is crucial for assessing model performance in practical scenarios, there is a trade-off between reproducibility, cost, safety, and real-world applicability. Simulated benchmarks offer a standardized, easily reproducible environment, reducing the time, financial cost, and safety risks for researchers to replicate results [1,2]. **In our revision, we will include a discussion about this limitation in the main paper.**
>
> [1] Evaluating Real-World Robot Manipulation Policies in Simulation. CoRL 2024.
>
> [2] Visualagentbench: Towards large multimodal models as visual foundation agents. ICLR 2025.
>
> **Q2: How to prevent failure of plans? Any qualitative examples?**
>
> **A2:** In EmbodiedBench, we allow MLLM agents to generate multiple actions at once. If a failure occurs (e.g., invalid actions or unmet goals), they **replan** using the latest image and interaction history. **In Appendix F, we provided four qualitative examples of our MLLM agents (Figures 13–16)**, demonstrating their ability to replan effectively.
>
> **Q3: Some results are repeated and can be (optionally) removed from appendix**
>
> **A3:** In Appendix D, we opt to include both the results from the main paper and additional results to provide a clearer trend across different tasks.
>
> **Q4: Using panoramic images for multi-view**
>
> **A4:** We included a panoramic image in our multi-view ablation for EB-Navigation. It provides a top-down perspective, capturing the entire scene. However, we found that it can mislead the agent, negatively impacting overall performance. Examples of the multi-view setup are shown in Figure 1 of https://anonymous.4open.science/r/rebuttal-3568/rebuttal_file.pdf.
>
> **Q5: Train policies on this benchmark and show performance ... this is not a necessity since the purpose of the benchmark is to evaluate MLLMs.**
>
> **A5:** We agree that further fine-tuning is a promising direction, but a key challenge is the lack of training data. Currently, the only available embodied planning dataset in our setting is ALFRED, which lacks the structured perception and reasoning. To address this, we are collecting trajectories using our agent framework on the base subset while reserving other subsets for evaluation. This will benefit future fine-tuning MLLM research, and we aim to release the dataset soon.
>
> **Q6: Difference between the complex instruction understanding and long-horizon tasks. Any intersection?**
>
> **A6:** The two subsets target different challenges:
> - "Complex Instruction" **adds longer, relevant/irrelevant context**, making user instructions harder to interpret while keeping task complexity similar to the base subset.
> - "Long Horizon" **increases task difficulty** by requiring more steps to complete.
>
> In EB-ALFRED, GPT-4o takes an average of 13.4 steps (base), 14.2 steps (complex instruction), and 23.9 steps (long-horizon), showing their differences. **We ensure no subset overlap in our benchmark design**.
>
> **Q7: How is the data generated for EB-Navigation? Which Python program was used? I did not find this in appendix.**
>
> **A7:** We describe the data generation process in **Appendix B.3** but will clarify it further in our revision. The EB-Navigation dataset consists of: 1. scene and object information, 2. initial robot position and pose, 3. target object information, 4. language instruction. We create the dataset using a Python program that ensures the validity of the (1,2,3,4) combinations. The process follows these steps:
> - Step 1: Scene and Object Initialization
> We use 90 scenes from AI2-THOR-supported scenes. Each scene initializes a set of objects.
> - Step 2: Target Object Selection
> In each scene, we iterate over potential target objects, excluding those inside receptacles or with multiple instances to reduce ambiguity.
> - Step 3: Agent Position and Pose Determination
> For each target object, we use AI2-THOR’s `GetInteractablePoses` to randomly sample a valid agent position, ensuring a distance at least 2.5 meters from the target. The agent's pose is set to either include the target object in view (e.g., base subset) or keep it out of view (long horizon).
> - Step 4: Instruction Generation and Augmentation
> Based on predefined templates (e.g., "Move towards the {target object} and stay near it"), we apply GPT-4o to augment linguistic diversity and preserve subset requirements.
>
> After executing the above data generation, we select 60 tasks for each subset to form the EB-Navigation dataset.
>
> **Q8: Typos**
>
> **A8:** Thank you for pointing out these typos. We have carefully corrected them in our revised manuscript.

---

> > ### Comment · Reviewer_yZa7 · 2025-04-09
> >
> > Thanks to the authors for rebuttal.
> > It addresses all of my concerns. It is interesting to see that the authors are already working on fine-tuning on the benchmark, looking forward to the results on those in a future work.
> >
> > Also, I like the panoramic multi-view top-down experiment. I am not sure if a top-down view is the best way, maybe an "ego-centric" panorama is a better choice? Regardless, this is just a suggestion and does not reduce the strength of this paper.
> >
> > I am raising the score to 5.

---

### Decision · Program_Chairs · 2025-05-01

**Decision:**

Accept (oral)

**Comment:**

The paper introduces EmbodiedBench, a benchmark designed to evaluate the capabilities of multi-modal large language models (MLLMs) in the context of embodied AI tasks. Different MLLMs are evaluated on this benchmark, providing valuable insights into how these models perform across various tasks.

Strengths:
The paper introduces a novel and valuable benchmark for evaluating MLLMs across a broad set of tasks, from high-level reasoning to low-level manipulation. Thorough evaluations across various environments are conducted, providing insights into the strengths and weaknesses of different MLLMs. It presents new findings, such as the importance of visual cues in low-level tasks and the challenges MLLMs face in long-term planning and complex instruction understanding.

Weaknesses:
The paper uses a single prompt across all models, which might influence model performance.

All reviewers are positive towards this paper, clearly an acceptance.